# Environmental Footprint of GenAI Research: Insights from the Moshi Foundation Model

## Abstract

New multi-modal large language models (MLLMs) are continuously being trained and deployed, following rapid development cycles. This generative AI frenzy is driving steady increases in energy consumption, greenhouse gas emissions, and a plethora of other environmental impacts linked to datacenter construction and hardware manufacturing. Mitigating the environmental consequences of GenAI remains challenging due to an overall lack of transparency by the main actors in the field. Even when the environmental impacts of specific models are mentioned, they are typically restricted to the carbon footprint of the final training run, omitting the research and development stages.

In this work, we explore the impact of GenAI research through a fine-grained analysis of the compute spent to create Moshi, a 7B-parameter speech-text foundation model for real-time dialogue developed by Kyutai, a leading privately funded open science AI lab. For the first time, our study dives into the anatomy of compute-intensive MLLM research, quantifying the GPU-time invested in specific model components and training phases, as well as early experimental stages, failed training runs, debugging, and ablation studies. Additionally, we assess the environmental impacts of creating Moshi from beginning to end using a life cycle assessment methodology: we quantify energy and water consumption, greenhouse gas emissions, and mineral resource depletion associated with the production and use of datacenter hardware.

Our detailed analysis allows us to provide actionable guidelines to reduce compute usage and environmental impacts of MLLM research, paving the way for more sustainable AI research.

## 1 Introduction

The environmental footprint of modern artificial intelligence systems has become a growing concern (Zhuk, 2023), as the rapid scaling of its energetic requirements unfolds on a planet already under significant ecological strain (Rockström et al., 2024). Large text and multimodal models now require millions of GPU-hours for training alone (Zhao et al., 2025; Le Scao et al., 2022), raising questions about the sustainability of current development practices (Varoquaux et al., 2025) and MLLM research itself.

In this work, we present a detailed analysis of the full development life cycle of Moshi, a state-of-the-art multimodal speech-text foundation model developed by Kyutai, a leading research organization in large language models and speech technologies. Rather than focusing exclusively on the final training run or hyperparameter search, we study the complete sequence of research activities that led to the released system, from early exploratory experimentation through final training.

This broader perspective is essential. Indeed, most research works in machine learning only report the cost of the training of the final model, implicitly assuming it to be the dominant contributor to overall cost. A few environmental impact studies have also considered the cost of hyperparameter search, and others the impact of inference, but research practices largely remain terra incognita. In practice, MLLM research involves extensive experimentation, debugging, hyperparameter tuning, architectural exploration, benchmarking, ablation studies, and discarded runs. These activities can collectively account for a substantial

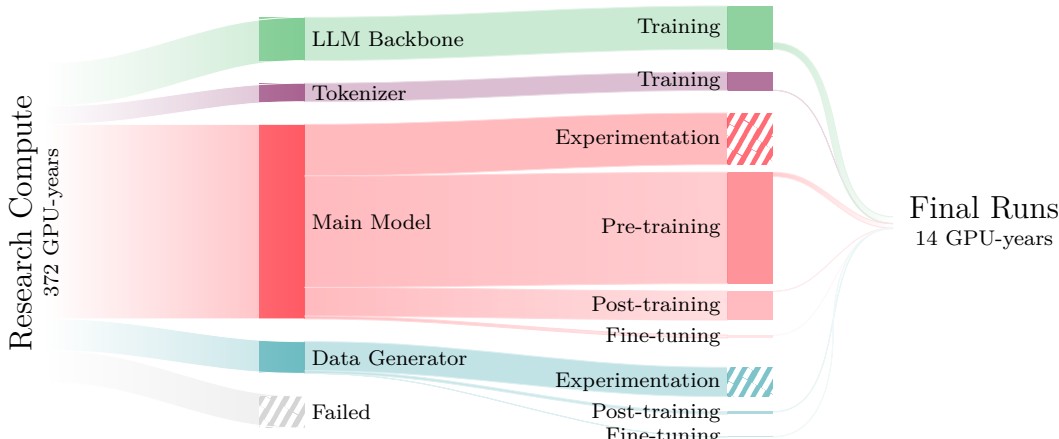

Figure 1: **From research to final compute.** *Research Compute* is split among individual model components and their respective training phases. *Failed* reflects the cost of failed experiments, and *Experimentation* gathers early versions that differ significantly from the definitive architecture and training scheme choices for specific components. *Final Runs* isolates the compute of training only one definitive version of each model component.

share of both compute usage and environmental impact, yet they are rarely measured or reported. Our study is enabled by an unprecedented level of access to internal training logs provided by Kyutai. This access allows us, for the first time, to quantify the computational and environmental impact of *all* stages of a specific industry-scale MLLM research project, beyond standard hyperparameter search.

In addition, we conduct a comprehensive life cycle assessment (LCA) of the Moshi research project. Beyond operational energy consumption, we estimate operational and embodied impacts in four impact categories.

Our analysis yields several key findings. First, we show that training the final deployed model accounts for only a small fraction of the total environmental impact, around 4%, while the environmental impact of experimentation, debugging, failed runs, model evaluation and ablation studies is significant. Second, we find that the ratio between the full cost and the final training highly depends on the novelty of the approach, being around 6.5× for the relatively standard LLM backbone but around 40× for the main Moshi model. Finally, we observe a strong decoupling between the number of runs and their intensity: 13% of runs account for nearly 89% of the total compute. Together, these results shed new light on current development workflows and highlight concrete opportunities to reduce the environmental footprint of future AI systems.

## 2 Related Work

We first review the core literature on LCA methodology and discuss the specific challenges of applying it to AI systems (sec. 2.1). We then survey prior work that applies these methodologies to AI systems (sec. 2.2). Finally, we provide an overview of Moshi, the system analyzed in this paper (sec. 2.3).

### 2.1 Methodologies for Environmental Assessment

**Life Cycle Assessment (LCA) Methodology.** Life cycle assessment (LCA) (Hauschild et al., 2018; Heijungs & Suh, 2002) is an environmental impact assessment methodology formalized in the 14040/44 ISO standards. It was proposed as a holistic approach to evaluating the environmental impacts incurred throughout the *life cycle* of a product system: raw material extraction, manufacturing, transport, use, and end of life. LCA considers a variety of *impact categories* with effects on human health, natural resources, and the natural environment, measured via *indicators* such as global warming potential, water consumption,

or human toxicity. These impacts are assigned with respect to a *functional unit*: a quantitative description of a function provided by the system at a desired level of performance. LCA may be *attributional*, aiming to quantify the environmental impacts that can be attributed to the product system; or *consequential*, aiming to predict the environmental consequences of introducing the product system on the market.

LCA is essential to diagnose impacts shifting between life cycle phases and impact categories. For example, using more efficient hardware reduces energy consumption, but increases production impacts due to semiconductor manufacturing.

**LCA Methodology for AI Systems** An AI system encompasses an AI model and the tangible infrastructure involved in its creation and deployment: sensors used for data collection, hyperscale datacenters for training, servers for hosting the model and running inference, etc. Initiatives for adapting the LCA methodology to AI systems have recently been proposed (Ligozat et al., 2022; OECD.AI Expert Group on AI Compute and Climate, 2022; Kaack et al., 2022), based on frameworks specific to information and communications technology (ICT) (Hilty & Hercheui, 2010; ITU, 2024). These developments have led to tools such as MLCA (Morand et al., 2024) or Boavizta (Simon et al., 2025), which we leverage in this work.

Several distinctions in the type of impact are important for our work. First, the impact can be attributed to three types of effects (Horner et al., 2016; Hilty & Hercheui, 2010; Kaack et al., 2022): (i) *first-order effects* are directly related to the development and operation of AI systems; (ii) *second-order effects* result from changes in industry when using an AI system; (iii) *third-order effects* are large-scale changes in lifestyle and economic structures following the widespread use of AI. Second, first-order impacts can be divided into *operational impacts* incurred directly while using the hardware, and *embodied impacts* corresponding to the remaining life cycle phases of the hardware, such as manufacturing, transport and end of life (Horner et al., 2016; Gupta et al., 2022; Kaack et al., 2022). Third, the *AI system development life cycle* (Wu et al., 2022) can be divided into four stages: (i) data collection, processing, and storage; (ii) research and development; (iii) model training; (iv) model deployment (inference). We highlight the distinction between *research*, which involves free-range experimentation on modeling choices, model architecture design, training techniques, and other aspects; and *development*, which entails more structured hyperparameter searches and scaling law experiments in preparation for final model training.

In this work, we adopt an attributional approach and consider the operational and embodied first-order impacts of the research and development and training stages of a speech-text AI foundation model. Detailing the impact of the research phase is the key novelty of our work compared to existing assessments, that we detail in the next section.

## 2.2 Environmental Assessments of AI Systems.

In this section, we give an overview of existing environmental reports for AI systems. We first list some works that assess the impact of AI services without considering the research and development stage. We then outline works that focus on research and development impacts and are closer to our study. Finally, we mention company reports on their LLM development costs, which are relevant but often remain very high-level.

**AI System Deployment.** As the use of commercial AI solutions has become widespread, a body of work has focused on assessing the growing impacts of model deployment, considering both embodied and operational impacts. Such works assess impacts ranging from just global warming potential (Gupta et al., 2022; Chien et al., 2023; Li et al., 2025b) up to several environmental impact indicators, including abiotic depletion potential (Berthelot et al., 2024) or water consumption (Elsworth et al., 2025). Jegham et al. (2025) extend their assessment of model deployment to three impact indicators, but do not consider embodied impacts. Opposite to these works, we focus on research and development costs.

**AI System Research and Development.** A few published studies assess the impacts of training suites of LLMs while also considering *development* overheads. Among these, some report the impact of development activities as a single number (Lakim et al., 2022), whereas others provide breakdowns by model size (Morrison et al., 2025) and training stage (Morrison et al., 2026).

Strubell et al. (2019) were arguably the first to quantify the *research and development* cost of a novel NLP model. In their work, they compare the impacts of training a single instance of the model, of tuning the model, and of the full research and development process. They also estimate the impacts of running a neural architecture search to improve model architecture, which was revisited by Patterson et al. (2021). Other more recent works also consider research costs, with varying levels of granularity: Wu et al. (2022) merge research, development, and training costs into a single "offline training" cost, whereas Luccioni et al. (2023) provide a breakdown by model size and high-level activity, i.e. model evaluation and miscellaneous processes. In contrast, we provide a detailed breakdown of research costs for a foundation model approaching a task in a radically novel way.

**Company Reports on LLM Development.** Well-known LLM developers have reported the environmental footprint of training specific models, but, to the best of our knowledge, none of these reports give detailed insights of the research and development process: Google and Meta AI estimate the final training carbon footprints of T5, GPT-3 (Patterson et al., 2021), Gemma (Gemma Team et al., 2024), the Llama family (Touvron et al., 2023a;b; Meta, 2024), OPT (Zhang et al., 2022), and other models, without considering research and development costs. The OPT model report (Zhang et al., 2022) provides a logbook registering informal comments made during development, but does not quantify the impact of the registered incidences. Similarly, the model report of Gopher (Rae et al., 2022), which focuses on compute, omits "compute arising from development, pre-emption, or other sources of inefficiency", although it does offer a detailed account of the compute spent on evaluation across several benchmarks.

These same entities do, however, quantify research and development costs at the company level: Wu et al. (2022) mention the share of infrastructure power capacity devoted to experimentation at Facebook AI, and indicate the compute intensity of typical experimental runs. On the other hand, Patterson et al. (2022) report the energy consumption spent on machine learning at Google -including research, development, testing, and production- but do not isolate the consumption of each of these activities.

Other companies are more transparent regarding their environmental impacts: Allen AI provides holistic assessments of the OLMo model family (Groeneveld et al., 2024; Team OLMo et al., 2025), quantifying the impacts of development training runs for different model sizes (Morrison et al., 2025) and training stages (Morrison et al., 2026); and Mistral reports the results of a comprehensive life cycle assessment of two of its LLMs (Mistral AI, 2025), but without clear mention of the research and development stage.

Contrary to these reports, we analyze in detail all the compute that was used in the research and development of the Moshi speech-text foundation model.

## 2.3 Background on Moshi

Moshi (Défossez et al., 2024) is an open-source speech-text multimodal foundation model designed for natural, expressive, and real-time interaction. It was developed by Kyutai[1] over a 9-month period and is arguably the first end-to-end speech-to-speech model, making it a good case study for assessing the impacts of innovative research at an industrial scale, which go beyond the hyperparameter tuning or dataset validation common in well-established LLM development pipelines.

Kyutai kept and shared detailed logs of the 3,534 individual training runs necessary for the development of the most innovative parts of their model, enabling detailed analysis of research costs. They also provided us with the global development and final training cost for their more standard LLM backbone.

As illustrated in fig. 2, the development of Moshi relies on the following four modules:

- **LLM backbone**: Helium, a pure-text LLM trained from scratch and used to initialize Moshi's main transformer.
- **Data generator**: a text-to-speech module used to create custom fine-tuning datasets.
- **Tokenizer**: Mimi, a neural audio codec that converts waveforms to speech tokens and back.

---

[1] https://kyutai.org/

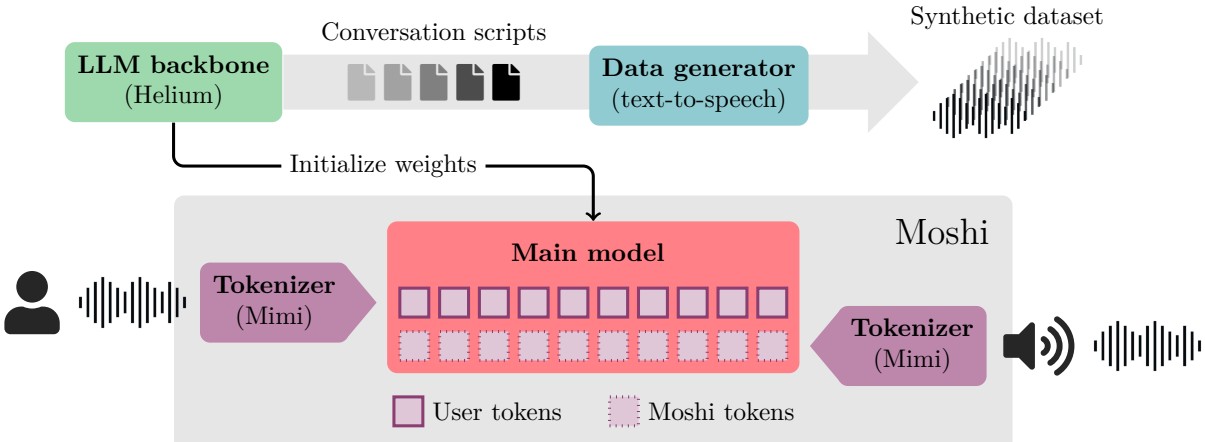

Figure 2: **Moshi modules.** Mimi ▇ tokenizes input waveforms and feeds them to the main transformer model ▇, whose predictions are converted back to waveform by Mimi. The transformer is initialized with the weights of the custom LLM Helium ▇, and a data generator ▇ converts synthetic conversation scripts into a fine-tuning speech dataset.

- **Main model**: a 7B-parameter transformer that consumes and produces tokenized speech.

As all the runs took place on identical compute nodes on the Scaleway cloud-computing platform[2], we define the *compute* associated to a run as its duration multiplied by the number of GPUs, and use it as the main unit of comparison in our analysis. For example, a run executed on 8 GPUs for 10 hours has a compute intensity of 80 GPU-hours.

## 3 Compute Distribution Analysis

In this section, we analyze how compute is allocated across high-level objectives, including training, evaluation, hyperparameter search, ablations, and final model training (sec. 3.1). We then examine Moshi in detail, quantifying the compute cost of its modules and training phases (sec. 3.2). Finally, we characterize the distribution of compute by analyzing the intensity of the training runs across these categories (sec. 3.3).

### 3.1 Compute Distribution by Goal

In this section, we first quantify the compute spent on each run phase (training, validation, and evaluation), and then analyze how compute is distributed across research phases (from debugging to ablation studies).

**Compute per Run Phase.** Within a run—the training of a module with a fixed hyperparameter configuration—we distinguish three phases:

- **optimization** per se, where gradients are computed through backpropagation, and the parameters are updated,

- **validation**, where, periodically during training, inference is performed on a held-out subset of the data to monitor optimization progress and detect overfitting,

- **evaluation**, where, periodically during training, inference is performed on the test set. Model outputs are saved for human assessment, and more metrics are computed, potentially requiring the generation of large volumes of data and scoring with external models.

---

[2]https://www.scaleway.com/

Training Run Phases

Research Phases

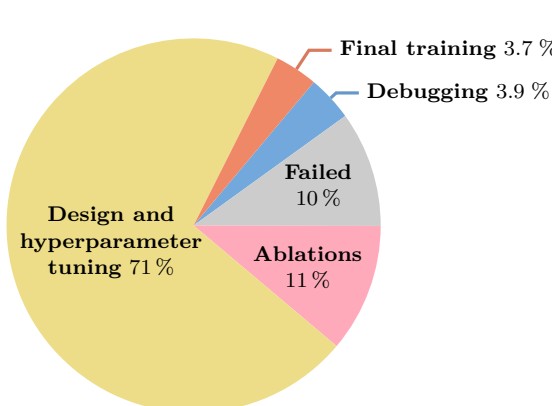

Figure 3: **Compute per run phase.** Runs are split into training, validation, and evaluation. We aggregate the compute for each phase across all runs, excluding LLM development.

Figure 4: **Compute per research phase.** AI research includes design, tuning, and final training, but also debugging, failed runs and ablations.

Fig. 3 shows the breakdown of compute along these three run phases. Core training accounts for 90% of the total compute and is by far the dominant computational driver, but validation and evaluation still account for 2.8% and 7.2% of the compute budget respectively, i.e., over 35 GPU-years, which is far from negligible. Note that approximately one quarter of the evaluation compute is dedicated to sample generation for human assessment.

**Compute per Research Phase.** We distribute the compute cost across the main research phases that we identify from the logs:

- **debugging**,

- **failed runs**, corresponding to unusable training runs that yielded very low performance and were therefore not used for hyperparameter selection,

- **core development, model design, and hyperparameter tuning**, including experimental exploration of directions that were later discarded,

- **final model training**,

- **ablation studies and safety analyses**, performed for rigorously validating design choices, writing the scientific paper, and making the model ready to be released.

As shown in fig. 4, 71% of the compute is devoted to architecture design and hyperparameter tuning, whereas training the final models accounts for less than 4% of the total compute. The ablation studies and safety analyses reported in Moshi's white paper alone represent 11% of the compute budget. Notably, debugging and failed runs together account for almost 14% of the total compute, underscoring their substantial contribution to overall resource usage.

> **Key Takeaways**
>
> - **Periodic evaluation during training adds a significant overhead**: over 7% of the compute is spent on evaluating models in case the need for deeper analysis or human assessment arises, a non-negligible share that calls into question the need for additional extensive online evaluation on top of validation.
> - **Final model training is a small part of the total research and development budget,** accounting for less than 4% of the total compute, emphasizing the importance of research costs.
> - **Debugging and failed runs are costly**: together, they account for almost 14% of compute, suggesting that improved debugging practices and early-phase experiment diagnostics could significantly reduce overall resource consumption.
> - **Ablation studies are costly**: while they are key to rigorous design choice validation and research publications, ablation studies represent 11% of the total computation budget, a considerable share that invites to consider alternative design validations.

### 3.2 Compute Distribution by Module and Training Phase.

**Training Phase Definition.**   In this section, we identify the training runs corresponding to each module of Moshi described in sec. 2.3. To better understand the research process, we also classify the runs of the most innovative parts of Moshi, namely the data generator and the main transformer model, into sequential training phases that could be applied to many AI development workflows:

- **Experimentation (Exp):** An exploratory phase used to test alternative architectures and functional modes; nothing is finalized at this point, and proof-of-concept models are evaluated through numerous runs, typically with moderate compute.
- **Pre-training (Pre):** Once the general pipeline and architecture type are fixed, models are trained on a large corpus. This phase typically involves fewer but substantially more compute-intensive runs. *Example:* Moshi learns speech representations from a dataset of 7M hours of audio, mostly containing English speech.
- **Post-training (Post):** The model's weights are refined to handle specific input/output formats using datasets tailored for this purpose. *Example:* Moshi learns conversational turn-taking, i.e., when to speak and when to listen, from a dialogue dataset with separate audio tracks.
- **Fine-tuning (FT):** The model is adapted to a specific application by training on smaller, specialized datasets. *Example:* Moshi learns to behave like a useful conversational assistant from 20k hours of synthetic instruction data.

For the more standard modules, namely the tokenizer and the LLM backbone, which do not require exploration and which are trained in a single phase, we simply refer to their training as **Train**. Note that this does not mean that a single training occurs, since more standard development activities such as hyperparameter search are still needed.

We also keep the **Failed** run category described in the previous section, common to all modules and training phases, and corresponding to runs that produced under-performing models due to factors such as bugs, misconfigured hyperparameters, or inadequate architectural variants.

A visualization of the development costs for the different phases can be seen in fig. 1, and we analyze them below in more detail.

**Project Timeline.**   In fig. 5, we visualize the number of runs and GPU in use (top) as well as the cumulative compute attributed to the different modules and training phases (bottom). It outlines that early phases of the project require a large number of short runs on few GPUs, and that the brunt of the computational load is taken by few expensive runs, especially long pre-training runs.

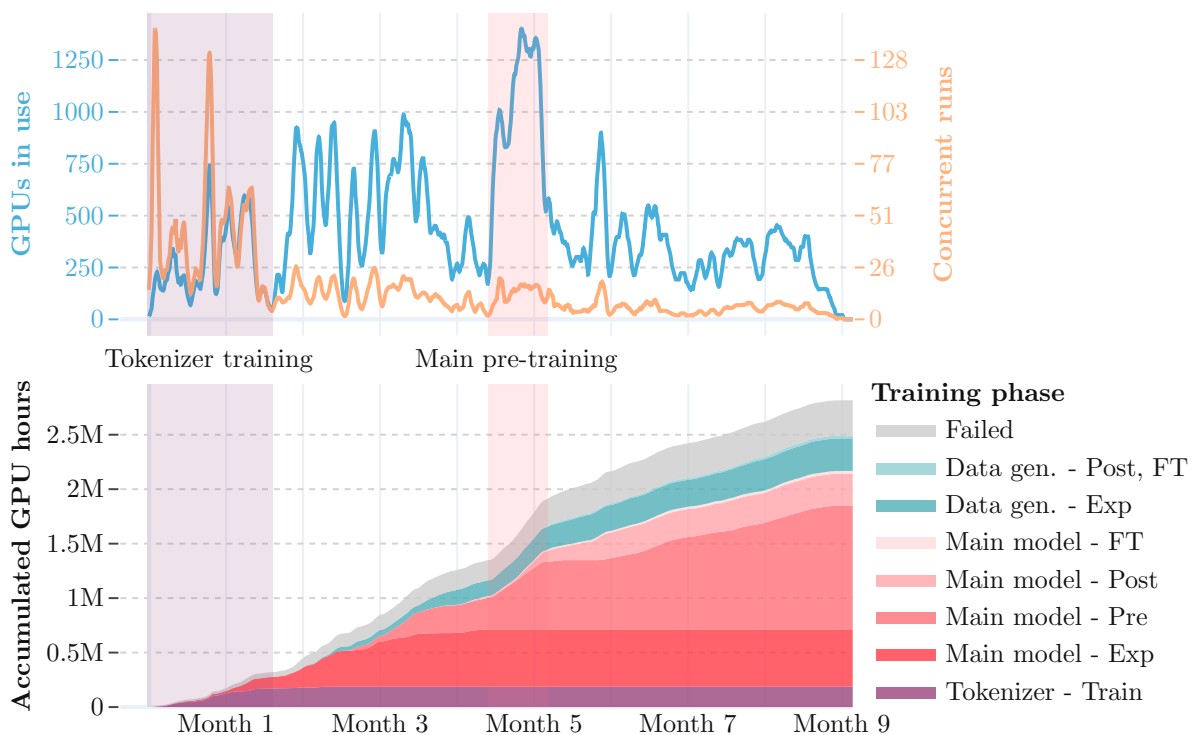

Figure 5: **Project compute intensity timeline.** *Top:* Number of GPUs in use (blue) and number of concurrent runs (orange) over the duration of the project. *Bottom:* Accumulated GPU hours per module and training phase: experimentation (Exp), pre-training (Pre), post-training (Post), and fine-tuning (FT). All quantities are sampled every 30 minutes, and smoothed with a sliding-window average over 100 steps. The plots do not include LLM backbone runs.

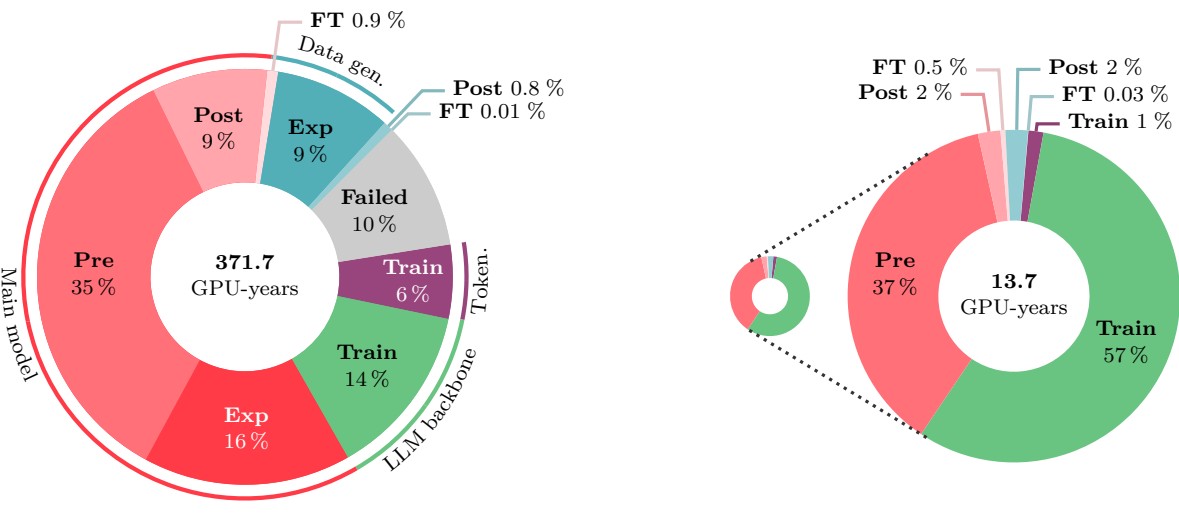

(a) **Research, development, and final runs.**  (b) **Final runs only.**

Figure 6: **Compute per training phase.** We distribute the compute among experimentation (Exp), pre-training (Pre), post-training (Post), and fine-tuning (FT) for each module. Fig. 6a considers all research and development runs plus the final training runs, and fig. 6b isolates the final runs. The area of each chart is proportional to the compute it represents.

Table 1: **Final vs. total compute.** We present the percentage of total research and development compute spent on the final training run, for each module and training phase independently: pre-training (Pre), post-training (Post), and fine-tuning (FT).

| Module | Tokenizer | Main model | | | Data generator | | LLM backbone |
|---|---|---|---|---|---|---|---|
| Training phase | Train | Pre | Post | FT | Post | FT | Train |
| Final-to-total compute ratio (%) | 0.9 | 3.9 | 0.9 | 2.0 | 10.6 | 8.2 | 15.5 |

**Distribution across Training Phases.** In fig. 6a, we report the distribution of the research and development compute between the training phases. Researching, developing, and training the main model dominates the budget (60%), with pre-training alone contributing 35%, while fine-tuning accounts for less than 1%. In contrast, developing and training the LLM represents 14% of the overall compute. Early experimentation is responsible for a significant share of compute, accounting for 25% of the total.

In fig. 6b, we report the same breakdown restricted to the final runs only, i.e. a single run for each training phase whose resulting model is deployed in production. Under this setting, training the LLM and pre-training the main model account for 57% and 37% of the final compute respectively, largely outweighing all other modules and training phases. The notable increase in the share of LLM compute is explained by the (proportionally) lower development costs of the more standard LLM backbone compared to other modules. On the contrary, the main model requires almost 60% of the total compute, but only 37% of the final training compute. We believe that this difference is actually related to the novelty of the different modules, the development cost of more standard modules being limited to hyperparameter search, while the most novel require more research and exploration.

**Relative Cost between Final Runs and Total Compute.** To better analyze this effect, we report in tab. 1 the ratios of the final training cost to the total research, development, and training cost for each module and training phase. The main model and tokenizer have the lowest ratios corresponding to high research and development costs. The case of the data generator is interesting: this module is in fact a variant of the main model, with slightly modified post-training and fine-tuning schemes. After designing and training the main model, training the data generator was thus easier, which results in higher ratios, still however below the more standard LLM.

> **Key Takeaways**
>
> - **Modules and phases whose final costs are negligible might have significant research costs**, emphasizing the importance of not only analyzing the final training and considering all training phases.
> - **Experimentation on novel elements has a significant cost, 25%**.
> - **The final-to-total cost ratios differ significantly by module**, the most standard module (LLM) having a much higher ratio, emphasizing the importance of assessing research costs.

### 3.3 Compute Distribution by Run Compute Intensity

We analyze the different runs according to their compute intensity, i.e., their GPU-time. We group runs into eight compute intensity categories with thresholds at one GPU-hour, one GPU-day, one GPU-week, one GPU-month, one GPU-year, three GPU-years, and five GPU-years.

**Compute Intensity of All Runs.** In fig. 7, we report both the number of runs per intensity category and their contribution to total compute. We observe a pronounced 90/10 effect, with a frontier around one GPU-month: 13% of runs account for 89% of the total compute. Low-intensity runs (below one GPU-day) represent 42% of all runs but contribute negligibly to total compute, as they are primarily associated with

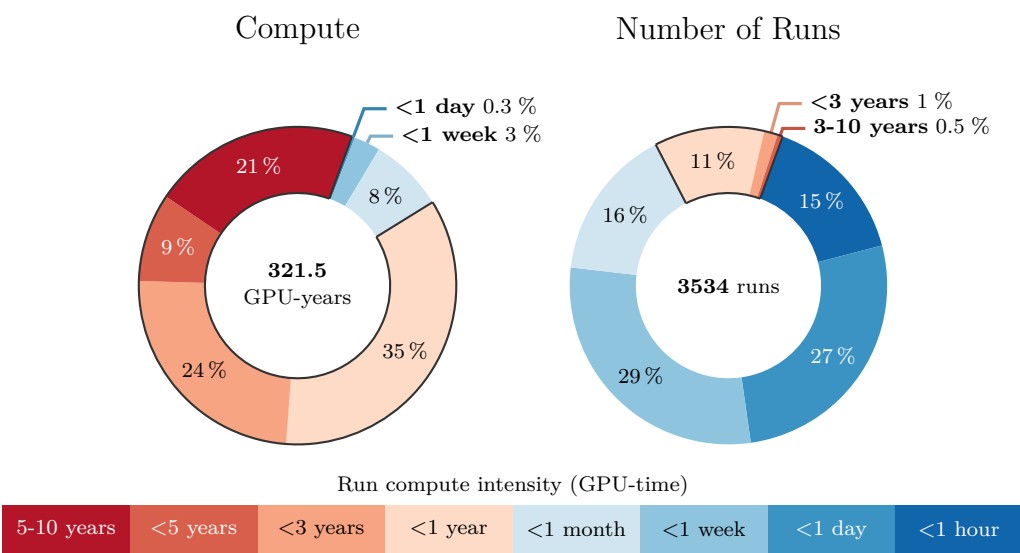

Figure 7: **Run compute intensity categories.** Distribution of runs across compute-intensity categories, excluding LLM runs. The highlighted sectors correspond to 90% of the compute concentrated in 10% of the runs.

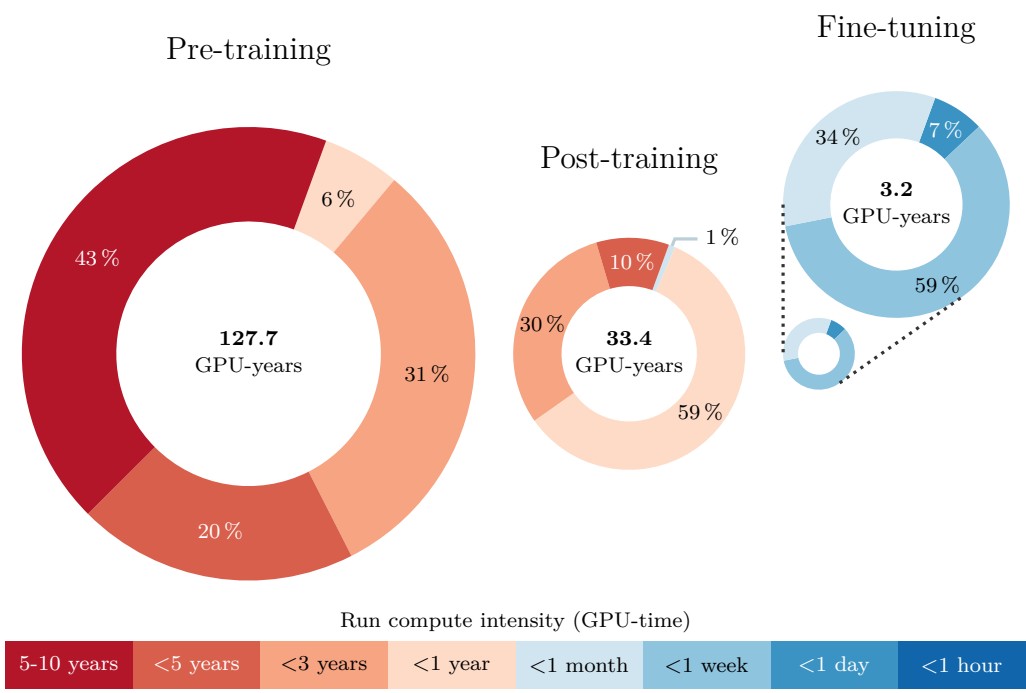

Figure 8: **Run compute intensity by training phase.** Compute-intensity distribution for pre-training, post-training, and fine-tuning runs of the main model. The area of each chart is proportional to the compute of the corresponding phase.

debugging, fine-tuning, and tokenizer training. At the opposite end of the spectrum, only 19 runs (0.5%) with intensities exceeding three GPU-years account for 30% of the total compute.

**Compute Intensity of Failed and Ablation Runs.** As observed in fig. 4, failed runs and ablation studies each contribute around one tenth of the development compute. However, failed runs are much less

compute-intensive than ablation studies: out of 1,472 failed runs, only four (0.3%) have an intensity over one GPU-year; whereas ablation studies only comprise eighty runs, thirteen of them (16%) over the one GPU-year mark.

**Run Compute Intensity by Training Phase.** We focus on runs from the pre-training, post-training, and fine-tuning phases of the main model and analyze their compute intensity (fig. 8). Pre-training concentrates the most compute-intensive runs and is the only phase containing runs exceeding five GPU-years, which make up 43% of the pre-training compute. In contrast, no fine-tuning run exceeds one GPU-month; instead, 66% of the fine-tuning compute corresponds to runs lasting under one GPU-week. Post-training occupies an intermediate regime, with 60% of its compute coming from runs under one GPU-year.

---

**Key Takeaways**

- **A small fraction of runs dominates compute usage**: 13% of runs account for nearly 90% of total compute.
- **Failed runs and ablation studies have distinct compute-intensity profiles**, with failed runs being less intensive, but ten times more frequent.
- **Pre-training concentrates the most compute-intensive runs**, including all individual runs exceeding five GPU-years.

---

## 4 Environmental Assessment

In this section, we quantify the environmental impacts of developing the Moshi model, from initial experimentation up to the final ablations. We estimate both operational impacts, arising from the electricity consumed during training, and embodied impacts, arising from the manufacturing of the hardware where the training was run, based on the total compute of the project. Sec. 4.1 details how compute is converted into environmental impacts.

More generally, embodied hardware impacts are directly proportional to compute (GPU-time), since they are allocated according to hardware use time. We assume the use time of hardware components other than GPUs to be proportional to GPU-time, as GPUs cannot operate independently from the rest of the compute node.

Operational impacts are less directly tied to computation, as they depend on the actual electricity consumption of each run. Since such measurements are unavailable retrospectively, we assume a constant average power consumption across runs using the utilization estimates provided by Kyutai's researchers. In practice, this means that the environmental impact distributions may not exactly match the compute distributions reported in sec. 3, although prior work suggests that the discrepancy remains limited (Morrison et al., 2025).

**Cluster Configuration.** All training runs took place on the Scaleway Nabuchodonosor supercomputer (Scaleway, 2024), an NVIDIA DGX SuperPOD (NVIDIA, 2025b) made up of 127 NVIDIA DGX H100 (NVIDIA, 2025a) nodes and located in Paris [3].

**Impact Indicators.** We estimate operational and embodied impacts across the following environmental impact indicators:

- **Primary energy (PE)** measures the consumption of renewable and non-renewable energy resources extracted from nature, expressed in megajoules (MJ) (Boavizta, 2023).
- **Global warming potential (GWP)** quantifies the contribution of greenhouse gas emissions to climate change, expressed in kilograms of carbon dioxide equivalent ($kgCO_2eq$) (Boavizta, 2023).

---

[3]Kyutai rented additional nodes in an unspecified location during a period of three months, but we omit this fact due to a lack of detailed data.

Table 2: **Life cycle assessment scope.** Impact indicators considered for each life cycle phase of the hardware. We group raw material extraction and manufacturing into a single *production* phase. (✔) means that the impacts are only partially accounted for (Simon et al., 2025).

| Impact indicator | Life cycle phase | | | |
| --- | --- | --- | --- | --- |
| | Production | Transport | Use | End of life |
| Primary energy (PE) | ✔ | (✔) | ✔ | |
| Global warming potential (GWP) | ✔ | (✔) | ✔ | |
| Water consumption (WC) | | | ✔ | |
| Abiotic depletion potential (ADP) | ✔ | ✔ | ✔ | |

- **Water consumption (WC)** measures the volume of water used and not returned to its original source (through evaporation, incorporation into products, or migration), expressed in liters (L) (Li et al., 2025a).

- **Abiotic resource depletion (ADP)** quantifies the depletion of non-renewable mineral and metal (ADPe) and fossil (ADPf) resources, expressed in kilograms of antimony equivalent (kgSbeq) (Boavizta, 2023).

**Scope.** The object of our assessment covers the complete research and development process of the Moshi model, from early experiments to final training runs and ablations. We exclude data acquisition, processing, and storage due to a lack of detailed information, and we do not account for the environmental costs associated with deployment and inference after public release. Specifically, the *functional unit* of our assessment is: "Develop and release publicly a full-duplex speech-to-speech foundation model with a latency of 200 milliseconds".

As summarized in tab. 2, we omit end-of-life impacts because of the general lack of reliable data (Ficher et al., 2025; Baldé et al., 2024). We further restrict water consumption estimates to the use phase only. Although studies on the water consumption of hardware manufacturing exist (Falk et al., 2025; Boyd, 2012) extrapolating these results to individual hardware components would require a level of expertise that is beyond the abilities of the authors. Nevertheless, it should be noted that, as shown in these and related works (Hess, 2024), the water footprint of semiconductor manufacturing is considerable.

### 4.1 Methodology

This section provides a high-level description of our methodology for estimating the environmental impacts of developing and training Moshi, starting from measured compute usage. We refer the reader sec. B for the exact equations and parameters. We distinguish between *operational impacts*, which correspond to the use phase of the hardware, and *embodied impacts*, which are associated with hardware production, transport, and end of life.

**Operational Impacts.** The main resource consumed during AI training is electricity, which indirectly results in water consumption through datacenter cooling. Additional operational impacts arise from the power plants generating this electricity.

We first estimate GPU electricity consumption as the product of compute (GPU-time, defined as run duration multiplied by the number of GPUs used, aggregated across all runs), the maximum rated GPU power, and a GPU utilization factor. We set this utilization factor to 95%, based on observations from Kyutai indicating near-full GPU utilization during most runs, while accounting for brief periods of non-GPU-intensive work. However, these observations may not hold uniformly across all run types, particularly during under-optimized early experimentation or debugging runs. As a result, assuming a constant utilization factor close to full utilization for all runs may overestimate electricity consumption. We therefore provide a sensitivity analysis of this parameter in sec. C.

We estimate the electricity consumption of CPUs, RAM, and the remaining node hardware in a similar manner, rescaling GPU-time as appropriate based on the quantity of hardware per compute node. Based on Kyutai's observations, we assume a CPU utilization of 5%. Following prior analyses of similar compute nodes (Spetko et al., 2020), we assume the power consumption of RAM and other node components, including fans, SSDs, network cards, and the motherboard, to remain constant during training runs.

We account for energy overheads associated with storage, management, and communication at the SuperPOD level, as well as datacenter infrastructure overheads. We do not include the consumption of idle compute nodes, assuming that nodes not used for developing Moshi were allocated to other workloads by Scaleway.

To compute primary energy impacts, we follow the methodology of Boavizta (2026) and multiply energy consumption by the consumption of fossil fuel resources per kilowatt-hour, and adding an overhead to also account for renewable energy sources. We similarly multiply by an abiotic depletion factor per kilowatt-hour to obtain use phase resource depletion impacts[4].

Following prior work (Luccioni et al., 2023; Lannelongue et al., 2021), we estimate greenhouse gas emissions as the product of total energy consumption and the yearly average carbon intensity of electricity at the location of computation. Finally, we estimate water consumption using the methodology proposed by Li et al. (2025a).

**Embodied Impacts.** Hardware manufacturing, transport, and end-of-life have direct impacts on the environment owing to rare mineral mining, ultrapure water consumption, fluorinated gas emissions, and more (Hess, 2024).

We first estimate the unitary embodied impacts of GPUs, CPUs, RAM, SSDs, power supplies, motherboards, and chassis, as well as the assembly of the compute nodes. We then allocate embodied impacts to our functional unit, using the compute (GPU-time) associated with Moshi's development relative to the total hours of use of the hardware throughout its lifespan, following established practice in prior work (Luccioni et al., 2023; Morand et al., 2024; Falk et al., 2025). For components other than GPUs, we rescale compute based on the quantity of hardware per compute node.

We estimate unitary embodied impacts using per-component impact factors provided by Boavizta (Simon et al., 2025). For GPU embodied impacts, we refer to a recent report by ADEME (Lees-Perasso et al., 2026).

To estimate the total hours of use of the hardware, we assume a hardware lifespan of four years, in line with values employed in related work (Morand et al., 2024; Schneider et al., 2025; Falk et al., 2025; Desroches et al., 2025), and a reasonable average utilization rate of 60% for all hardware components (Luccioni et al., 2023; Wu et al., 2022). We provide a sensitivity analysis of both parameters in sec. C.

## 4.2 Analysis

We first report the total environmental impacts across the four indicators under study, comparing them, when possible, to yearly per-capita impacts. We then disaggregate each indicator by hardware component and by scope (computation, datacenter overheads, and embodied impacts), before focusing specifically on hardware production. Finally, we conclude with a simulation of how the environmental impacts of model development would vary across different geographic locations. Detailed numerical results are reported in sec. A.

**Environmental Impacts of Research.** The research, development, and training compute of the model ascended to 3M GPU-hours, or **372 GPU-years**. This translates into:

- An energy consumption of **5 gigawatt-hours**, equivalent to the yearly consumption of 727 people in France (RTE, 2025) and resulting in **68 terajoules** of primary energy extracted from the environment.

---

[4]Like Boavizta, we only contemplate mineral and metal resource depletion (ADPe) in the use phase, excluding fossil resource depletion (ADPf).

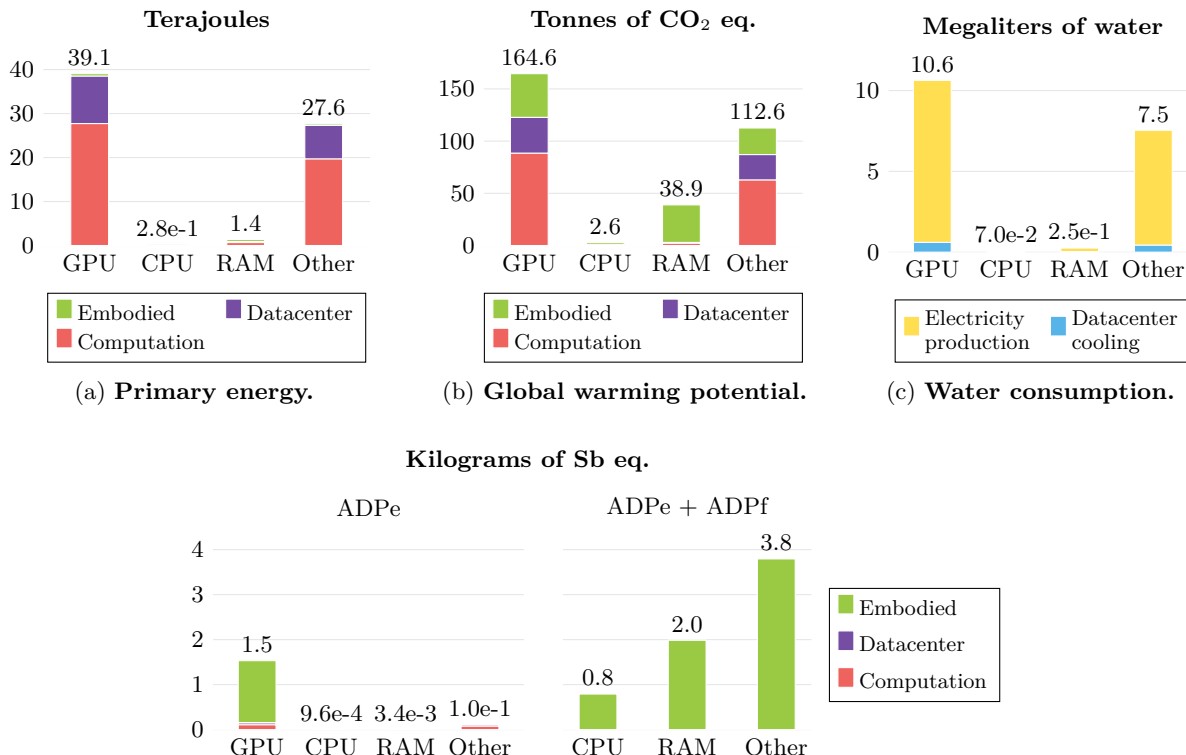

(d) **Abiotic depletion potential.** Depletion of minerals and metals (ADPe) and fossil resources (ADPf).

Figure 9: **Environmental impacts of research.** Each impact indicator (primary energy, global warming potential, water consumption, abiotic depletion potential) is disaggregated by hardware component (GPU, CPU, RAM, Other), and by scope.

- A global warming potential of **319 tonnes of carbon dioxide equivalent**, that of 39 people in a year in France (Baude & Larrieu, 2025), or of 132 round trip flights between Paris and San Francisco (Sustainable Travel International, 2024).

- A water consumption of **19 megaliters**, that of 342 people in a year in France (OECD, 2025).

- An abiotic depletion potential of **8 kilograms of antimony equivalent**, that of 6,566 smartphones (Sánchez et al., 2024) or 483 laptops (Baur et al., 2023).

Fig. 9 breaks down the total environmental impacts by hardware component and by impact scope: *computation* ■ due to compute node operation; *datacenter* ■ due to datacenter management, cooling, ventilation, and other overheads; and *embodied* ■ due to hardware production. For water consumption (fig. 9c), operational impacts are split between *datacenter cooling* ■ and power-plant cooling for *electricity production* ■.

When it comes to primary energy (fig. 9a) and global warming potential (fig. 9b), operational impacts due to computation ■ and datacenter overheads ■ are directly proportional. In contrast, embodied ■ impacts make up a larger share of the total global warming potential, whereas their contribution to total primary energy is small. This difference is very pronounced in the case of RAM. These larger embodied global warming potential impacts are mainly explained by the emission of fluorinated gases and wet chemicals during the manufacturing process (Hess & Nowicka, 2026).

Focusing on water consumption (fig. 9c), datacenter cooling ■ consumes a negligible amount of water in comparison to cooling the power plants that generate the electricity to power the datacenter ■. We do not estimate embodied water consumption, but it should be considerable due to the ultrapure water and the electricity required for semiconductor manufacturing (Boyd, 2012; Li et al., 2025a; Hess, 2024).

**Embodied impacts by component**

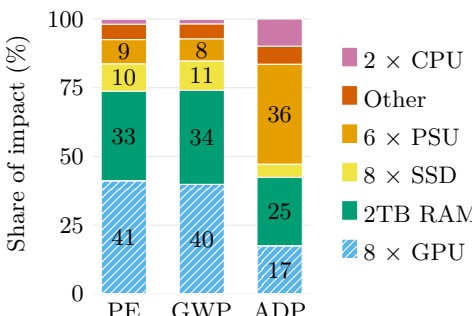

**Impacts by training location**

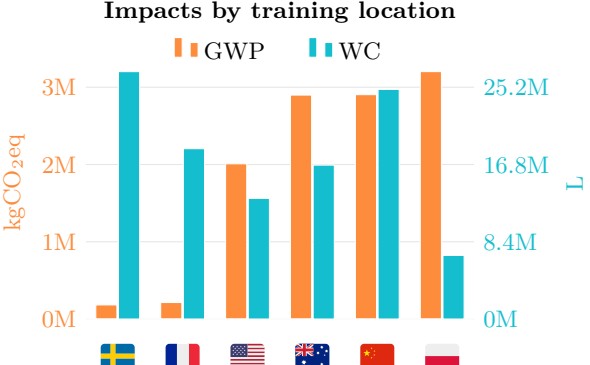

Figure 10: **Embodied impacts by component.** Share of embodied primary energy (PE), global warming potential (GWP), and abiotic depletion potential (ADP) for each hardware component in a node. Solid fill impacts are estimated using Boavizta (Simon et al., 2025) and include ADPe + ADPf; dashed impacts come from ADEME (Lees-Perasso et al., 2026) and include ADPe only.

Figure 11: **Operational impacts by training location.** Hypothetical global warming potential (GWP) and water consumption (WC) of developing the model in different locations, excluding embodied impacts.

GPUs are responsible for the majority of the impact across all indicators except abiotic depletion potential (fig. 9d), where node components other than CPUs and GPUs require the most material resources by a large margin. Notably, these other node components, including fans and network cards, contribute significantly to primary energy (fig. 9a) and global warming potential (fig. 9b), yet they are seldom taken into account in related work. Conversely, CPUs have the lowest impacts overall, which is explained by their low utilization during training.

**Embodied Impact Details.** We now break down embodied impacts for each impact category across the hardware components of a compute node. As shown in fig. 10, GPUs contribute the most to embodied primary energy (PE) and global warming potential (GWP), closely followed by RAM. The impacts of producing one RAM module are around five times lower than those of a GPU, but each compute node contains thirty-two RAM modules as opposed to only eight GPUs. The distribution changes for abiotic depletion potential (ADP), with power supplies being responsible for most of the resource depletion, followed by RAM modules and GPUs. The impacts of producing a single power supply are the highest among all components, and there are six power supplies in each compute node. We provide details on the per-unit impacts of each hardware component in tab. 5 and fig. 12 (sec. A).

**Importance of Location.** Factors such as carbon intensity and water consumption for power plant cooling vary by location, depending on the fuel mix powering the electricity grid. We explore the hypothetical operational impacts of developing Moshi in different locations in fig. 11. Overall, the figure shows no correlation between global warming potential and water consumption impacts; however, there is a marked trade-off between both impacts in Sweden, France, and Poland.

Carbon intensity in Sweden and France is low thanks to their reliance on hydroelectric and nuclear energy, as opposed to fossil fuels. However, hydropower and nuclear are among the most water-intensive energy sources (Reig et al., 2020). Conversely, Poland has one of the most carbon-intensive electricity grids worldwide (Electricity Maps, 2026), yet its water consumption is low: the Polish grid relies extensively on coal and gas, with almost no hydropower or nuclear power plants (EMBER, 2025b).

## 5    Discussion

We conclude this study by discussing our main results, how they compare to those available in the literature, and good practices to adopt in GenAI research. Even though our results are specific to this case study and therefore cannot be directly generalized to any other AI research projects, the research stages we discuss (debugging, ablation studies, evaluation, etc.)  are common to most AI development pipelines, and our findings suggest potential improvements of research practices.

**Final vs. Total Development Costs.**    The *final compute* of Moshi aggregates the GPU-time required to train, in successive phases, the released versions of each of its modules, whereas *total compute* represents the total compute of the project from beginning to end, including the final compute.

The final training of Moshi represents less than 4% of the total compute and estimated environmental impacts. This is much lower than the numbers reported in the literature. For example, according to Luccioni et al. (2023), training the final BLOOM model represented 31% of the compute (considering only a part of the project, the one performed on the cluster used for the final training), and Morrison et al. (2025) report that training the open source versions of the OLMo suite of LLMs accounted for 50% of the environmental impacts of the project. A more recent report on the latest generation of OLMo models (Morrison et al., 2026) places final run costs at 18% of the total compute of the project, with individual training phases reaching figures as low as 2%. We believe that this discrepancy is due to three main effects. First, while developed by experienced researchers, Moshi was built completely from scratch, following the creation of Kyutai, which we believe enables us to better account for the exploratory research phase than previous studies. Indeed, we argue that the boundary of a specific development project in institutes that have already performed LLM research is more blurry, as previous or related projects are likely leveraged. Second, we believe that this large difference is also due to the originality of the speech-to-speech Moshi model. While large language models benefit from years of empirical optimization—such as scaling laws (Hoffmann et al., 2022; Bhagia et al., 2025), which enable controlled experimentation on smaller proxy models, speech-to-speech modeling remains comparatively underexplored. As a result, a larger portion of the development process must be carried out at scale, increasing the relative cost of experimentation. This is reflected in our results: when restricting the analysis to the LLM backbone alone, the share of compute attributed to final training rises to 15%, bringing it closer to previously reported values. Third, a substantial fraction of the total compute is spent on experimentation, debugging, and other development stages that are often not accounted for in prior work.

**Reducing Unnecessary Compute.**    Failed experiments make up 10% of Moshi's total compute. Most of these training runs were quickly canceled, but their contribution is related to their large number: 42% of the runs were discarded due to poor hyperparameter combinations, mistaken configurations, or bugs. While failed experiments are unavoidable in research, these figures invite to take special care when launching compute-intensive experiments, e.g. above one GPU-month, and monitor them closely.

Debug runs, which are numerous but inexpensive individually, account for 3.9% (15 GPU-years) of the research and development compute in our case study, a share comparable to that of training the final model. Debugging is clearly essential, in particular because it can reduce overall compute usage by preventing failed large-scale training runs. Still, as our analysis shows, debug runs can accumulate into a non-negligible contribution. Researchers should therefore favor debugging on downscaled models and datasets whenever possible, and preferably use lower-power infrastructures rather than production environments. In most cases, debugging does not require access to frontier-scale hardware.

Likewise, periodic evaluation and validation during training account for 10% of Moshi's total compute, showing that performance tracking can induce a non-negligible computational cost. Periodic tracking during training can be important and has the potential to reduce overall compute usage by, for example, enabling early stopping or the diagnosis of unstable training runs, which can then be promptly terminated. Thus, considering the trade-off between potential benefits and costs of periodic evaluation, and limiting the compute spent on evaluation (e.g. keeping it under a small percentage of the compute of a run), seems important.

In many settings, this cost could be reduced by lowering evaluation frequency and using smaller validation sets.

**Questioning Research Practices and Expectations.** Ablation studies are often central to machine learning research articles, as they validate findings and methodological claims. However, they can also induce substantial computational costs: in our case study, 11% of Moshi's total compute was spent on ablation studies and safety analyses, mostly after the final model had already been trained. This share is roughly three times larger than the compute required to train the final deployed model itself, and stems primarily from expensive pre-training ablations.

We believe these findings encourage giving more care to ablation selection and design. For example, most comparisons could be carried out on smaller versions of the models and datasets or after much fewer training iterations. This also encourages an evolution of reviewers' mindsets, who should likewise remain mindful of the computational costs associated with expected ablations or analyses.

In a similar vein, we argue that given the strong environmental impact of AI research, the computational budget should be more systematically included in the evaluation, for example, evaluating expected performance as a function of the computational budget, including hyperparameter search (Dodge et al., 2019). Beyond reducing environmental impact, we believe that such considerations would also make comparisons more meaningful by factoring out the important impact of compute scaling on results (Mertens et al., 2026).

**Reducing Environmental Impacts for a Given Compute Budget.** The location of the computational resources has a significant effect on operational impacts. A natural direction to reduce AI research impact is thus to select data centers based on their power and water usage, and taking into account the carbon and water intensities of the local electrical grid. However, low-carbon grids might consume large amounts of water for power plant cooling, which is problematic in regions under water stress, and adapting the computational load to local resources demand is still a rare practice.

The largest share of embodied impacts stems from GPU and RAM manufacturing, as well as power supply production in the case of resource depletion. The impacts of GPU manufacturing are considerable due to both their high per-unit impact and the amount of GPUs required for training. Although producing a single RAM module has a lower impact, compute nodes designed for AI training may easily contain up to thirty or sixty of these modules. These observations should serve as an incentive to boost research into smaller models and training schemes with low memory footprints, as well as to extend the lifetimes of GPUs by reducing compute usage.

**Measurements and Publicity.** As a first step toward these evolutions and to better question the impact of GenAI, we argue that measuring and publicly reporting the computational and environmental costs of not only the final training, but also development, and even complete research projects, with breakdown per project stage, should become common practice. For operational impacts, tools such as CodeCarbon[5] are both easily accessible and accurate. Similar tools are also available for embodied impacts, for example, Boavizta (Simon et al., 2025) or MLCA (Morand et al., 2024), whose methodology we outline in sec. B.2.

---

[5]https://codecarbon.io

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

# A   Additional Results

In this appendix, we gather the numerical values represented in fig. 9 and fig. 11 of the main text: tab. 3 compiles all the results of our environmental impact assessment, disaggregated by hardware component and impact scope; and tab. 4 summarizes the hypothetical global warming potential and water consumption impacts of developing Moshi in different locations.

In fig. 10 of the main text, we show how the embodied impacts of a compute node are distributed across component types (GPUs, CPUs, RAM modules, etc.). We compliment this information in tab. 5 and fig. 12, which illustrate the impacts of producing *a single unit* of each component type, plus the assembly process of the full compute node.

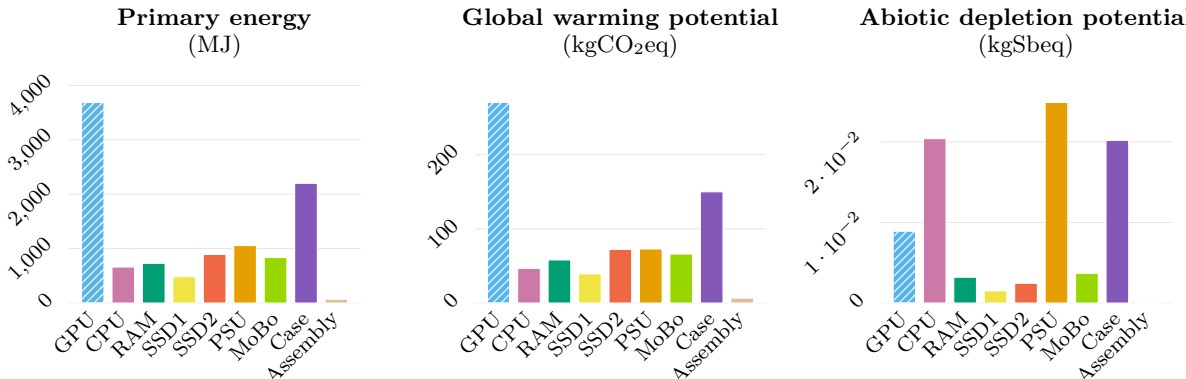

Figure 12: **Embodied impacts of one component.** Production impacts for *a single unit* of each hardware component. Solid fill impacts are estimated using Boavizta (Simon et al., 2025) and combine mineral and metal (ADPe) and fossil resource (ADPf) depletion in the case of abiotic depletion potential (ADP); dashed impacts come from ADEME (Lees-Perasso et al., 2026) and include ADPe only. We abbreviate *Motherboard* as *MoBo*. Values in tab. 5.

# B   Environmental Assessment Methodology

This appendix details all the formulas and values that we use to carry out our environmental assessment, thus providing complete transparency of our results and outlining sources of uncertainty. Sec. B.1 and sec. B.2 present the formulas used to derive, respectively, operational and embodied impacts.

## B.1   Operational Impacts

We employ the following methodology to estimate operational primary energy, global warming potential, and water consumption, using the values provided in tab. 6 and starting from the energy consumed by the datacenter:

Table 3: **Environmental impacts of research.** For each environmental impact indicator, we estimate embodied impacts due to hardware production and operational impacts due to computation and datacenter energy consumption overheads. In the case of water consumption, we discern between datacenter cooling and power plant cooling for electricity production. We abbreviate *Motherboard* as *MoBo*. Represented in fig. 9.

**Primary energy (MJ)**

| Scope | Component | | | | | | | | | Total |
| | GPU | CPU | RAM | SSD1 | SSD2 | PSU | MoBo | Case | Assembly | |
|---|---|---|---|---|---|---|---|---|---|---|
| Embodied | 5.71e5 | 2.56e4 | 4.52e5 | 1.87e4 | 1.38e5 | 1.23e5 | 1.62e4 | 4.26e4 | 1.33e3 | 1.39e6 |
| Datacenter | 1.08e7 | 7.09e4 | 2.55e5 | | | | 7.64e6 | | | 1.87e7 |
| Computation | 2.77e7 | 1.82e5 | 6.56e5 | | | | 1.97e7 | | | 4.82e7 |
| **Total** | 3.91e7 | 2.79e5 | 1.36e6 | | | | 2.76e7 | | | 6.83e7 |

**Global warming potential (kgCO$_2$eq)**

| Scope | Component | | | | | | | | | Total |
| | GPU | CPU | RAM | SSD1 | SSD2 | PSU | MoBo | Case | Assembly | |
|---|---|---|---|---|---|---|---|---|---|---|
| Embodied | 4.19e4 | 1.81e3 | 3.60e4 | 1.52e3 | 1.12e4 | 8.47e3 | 1.28e3 | 2.90e3 | 1.29e2 | 1.05e5 |
| Datacenter | 3.44e4 | 2.26e2 | 8.14e2 | | | | 2.44e4 | | | 5.98e4 |
| Computation | 8.83e4 | 5.81e2 | 2.09e3 | | | | 6.27e4 | | | 1.54e5 |
| **Total** | 1.65e5 | 2.62e3 | 3.89e4 | | | | 1.13e5 | | | 3.19e5 |

**Abiotic depletion potential (kgSbeq)**

| Scope | Component | | | | | | | | | Total |
| | GPU | CPU | RAM | SSD1 | SSD2 | PSU | MoBo | Case | Assembly | |
|---|---|---|---|---|---|---|---|---|---|---|
| Embodied | 1.38e0 | 7.90e-1 | 1.99e0 | 5.84e-2 | 3.80e-1 | 2.89e0 | 7.14e-2 | 3.91e-1 | 2.73e-05 | 7.95e0 |
| Computation | 1.05e-1 | 6.92e-4 | 2.49e-3 | | | | 7.46e-2 | | | 1.83e-1 |
| Datacenter | 4.09e-2 | 2.69e-4 | 9.69e-4 | | | | 2.90e-2 | | | 7.12e-2 |
| **Total** | 1.53e0 | 7.91e-01 | 1.99e0 | | | | 3.90e0 | | | 8.21e0 |

**Water consumption (L)**

| Scope | Component | | | | Total |
| | GPU | CPU | RAM | Other | |
|---|---|---|---|---|---|
| Datacenter cooling | 1.00e7 | 6.60e4 | 2.38e5 | 7.12e6 | 1.75e7 |
| Electricity production | 6.02e5 | 3.96e3 | 1.42e4 | 4.27e5 | 1.05e6 |
| **Total** | 1.06e7 | 7.00e4 | 2.52e5 | 7.54e6 | 1.85e7 |

**Energy Consumption.** We estimate operational energy consumption as the sum of energy consumed for computation ($\text{oper}_\text{E}^\text{computation}$) and datacenter overheads ($\text{oper}_\text{E}^\text{datacenter}$). We consider cluster management, networking, and storage overheads ($o_\text{cluster}$) (NVIDIA, 2025c); as well as cooling, ventilation, and other datacenter overheads (power usage effectiveness or PUE):

$$\text{oper}_\text{E} = \text{oper}_\text{E}^\text{computation} + \underbrace{((\text{PUE} - 1) \times o_\text{cluster} + (o_\text{cluster} - 1)) \times \text{oper}_\text{E}^\text{computation}}_{\text{oper}_\text{E}^\text{datacenter}}.$$

Energy consumption for computation is the aggregate of the consumption of all hardware components in a compute node:

$$\text{oper}_\text{E}^\text{computation} = \text{oper}_\text{E}^\text{GPU} + \text{oper}_\text{E}^\text{CPU} + \text{oper}_\text{E}^\text{RAM} + \text{oper}_\text{E}^\text{other},$$

Table 4: **Operational impacts by training location.** Hypothetical global warming potential and water consumption of developing the model in different locations, excluding embodied impacts. Represented in fig. 11.

| Operational impact | Location | | | | | |
|---|---|---|---|---|---|---|
| | Sweden | France | USA | Australia | China | Poland |
| Global warming potential (kgCO$_2$eq) | 1.83e5 | 2.13e5 | 2.01e6 | 2.90e6 | 2.90e6 | 3.20e6 |
| Water consumption (L) | 2.69e7 | 1.85e7 | 1.31e7 | 1.67e7 | 2.49e7 | 6.91e6 |

Table 5: **Embodied impacts of one component.** Production impacts for *a single unit* of each hardware component, plus the assembly process of the compute node: primary energy (PE), global warming potential (GWP), and abiotic depletion potential (ADP). We abbreviate *Motherboard* as *MoBo*. Represented in fig. 12.

| Impact indicator | Component | | | | | | | | |
|---|---|---|---|---|---|---|---|---|---|
| | GPU | CPU | RAM | SSD1 | SSD2 | PSU | MoBo | Case | Assembly |
| PE (MJ) | 3.69e3 | 6.62e2 | 7.30e2 | 4.83e2 | 8.93e2 | 1.06e3 | 8.36e2 | 2.20e3 | 6.86e1 |
| GWP (kgCO$_2$eq) | 2.70e2 | 4.67e1 | 5.81e1 | 3.93e1 | 7.23e1 | 7.29e1 | 6.61e1 | 1.50e2 | 6.68e0 |
| ADP (kgSbeq) | 8.94e-3 | 2.04e-2 | 3.20e-3 | 1.51e-3 | 2.45e-3 | 2.49e-2 | 3.69e-3 | 2.02e-2 | 1.41e-6 |

where the consumption for each hardware component is estimated as:

$$\text{oper}_E^{\text{hw}} = \mathcal{C} \times \frac{q_{\text{hw}}}{q_{\text{GPU}}} \times \begin{cases} u_{\text{hw}} \times \text{TDP}_{\text{hw}} & \text{if hw} \in \{\text{GPU}, \text{CPU}\} \\ P_{\text{hw}} & \text{if hw} \in \{\text{RAM}, \text{other}\} \end{cases},$$

with $\mathcal{C}$ the total development compute in GPU-hours, $q_{\text{hw}}$ the quantity of component *hw* per node, $u_{\text{hw}}$ the hardware utilization, $\text{TDP}_{\text{hw}}$ the thermal design power of the hardware, and $P_{\text{hw}}$ the constant power consumption of the hardware.

We establish average utilization factors for CPU and GPU ($u_{\text{CPU}}$ and $u_{\text{GPU}}$) based on Kyutai's observations during training.

To obtain the power consumption of RAM, we apply CodeCarbon's methodology with efficiency scaling (CodeCarbon, 2026), which results in $5 \times (4 + 4 \times 0.9 + 8 \times 0.8 + 16 \times 0.7) = 126$ watts for the 32 RAM modules of a compute node, and therefore 3.94 watts per RAM module ($P_{\text{RAM}}$). We define the power consumption of the remaining node hardware, $P_{\text{other}}$, as the difference between the total power consumption of a compute node (10,200 watts (NVIDIA, 2025a)) and the consumption of GPUs, CPUs, and RAM.

**Primary Energy (PE).** We compute operational primary energy as done in Boavizta (2026), starting from energy consumption:

$$\text{oper}_{\text{PE}} = \text{PE}_{\text{kWh}} \times \text{oper}_E,$$

with

$$\text{PE}_{\text{kWh}} = \frac{\text{ADPf}_{\text{kWh}}}{1 - \%\text{renewable}},$$

where $\text{ADPf}_{\text{kWh}}$ is the fossil resource depletion per kilowatt-hour of the electrical grid, and %renewable is the percentage of electricity generated from renewable energy sources.

**Global Warming Potential (GWP).** We estimate operational global warming potential by multiplying the operational energy consumption by the carbon intensity (CI) of the electrical grid:

$$\text{oper}_{\text{GWP}} = \text{oper}_E \times \text{CI},$$

making the appropriate unit conversions. Although not shown in the equation, we split operational global warming potential into impacts due to computation and datacenter overheads, as we do with primary energy.

**Abiotic Depletion Potential (ADP).** Similarly to primary energy, we compute operational mineral and metal resource depletion from energy consumption as follows:

$$\text{oper}_{\text{ADP}} = \text{ADPe}_{\text{kWh}} \times \text{oper}_{\text{E}},$$

with $\text{ADPe}_{\text{kWh}}$ the mineral and metal resource depletion per generated kilowatt-hour.

**Water Consumption (WC).** Following the methodology of Li et al. (2025a), we estimate operational water consumption as:

$$\text{oper}_{\text{WC}} = \underbrace{\text{WUE} \times o_{cluster} \times \text{oper}_{\text{E}}^{\text{computation}}}_{\text{oper}_{\text{WC}}^{\text{datacenter cooling}}} + \underbrace{\text{EWIF} \times \text{oper}_{\text{E}}}_{\text{oper}_{\text{WC}}^{\text{electricity production}}} \quad ,$$

where WUE is the water usage effectiveness of the datacenter, and EWIF is the electricity water intensity factor of the local electrical grid. Once again, we make the appropriate unit conversions.

We obtain the electricity water intensity factor (EWIF) as a weighted average of the water consumption per energy source reported by Reig et al. (2020), weighted by the energy mix in the target location in 2024 as reported in the EMBER database (EMBER, 2025b). Since Reig et al. (2020) and EMBER (2025b) do not use the same terminology to name energy sources, we establish correspondences as follows: *solar - photovoltaic*, *bioenergy - biomass*, *other renewables - geothermal*[6], *gas - natural gas*, *coal - hard coal*, and *other fossil - heavy fuel oil*. It should be noted that Reig et al. (2020) consider the water consumption of solar and wind energy to be zero, whereas other sources do not (Jin et al., 2019). For future work, we recommend referring to platforms such as Wattnet (Melguizo et al., 2026).

## B.2 Embodied Impacts

We estimate the production impacts for a single unit of hardware as follows:

$$\text{emb}_{\text{imp}}^{\text{CPU unit}} = \text{base}_{\text{imp}}^{\text{CPU}} + \text{die\_size}^{\text{CPU}} \times \text{die}_{\text{imp}}^{\text{CPU}}$$

$$\text{emb}_{\text{imp}}^{\text{mem unit}} = \text{base}_{\text{imp}}^{\text{mem}} + \frac{\text{capacity}^{\text{mem}}}{\text{density}^{\text{mem}}} \times \text{die}_{\text{imp}}^{\text{mem}} \ \forall \text{mem} \in \{\text{RAM}, \text{SSD}\}$$

$$\text{emb}_{\text{imp}}^{\text{PSU unit}} = \text{weight}^{\text{PSU}} \times \text{base}_{\text{imp}}^{\text{PSU}},$$

where $\text{imp} \in \{\text{PE}, \text{GWP}, \text{ADP}\}$ for primary energy, global warming potential, and abiotic depletion potential respectively. The impacts of the remaining hardware, and of the assembly process of the compute node, are constant values from Boavizta (Simon et al., 2025) and the ADEME report on GPU production impacts (Lees-Perasso et al., 2026). The base and die impact factors that we use are gathered in tab. 7, and tab. 8 lists the specifications of each hardware component. The allocated embodied impact for the duration of use is:

$$\text{emb}_{\text{imp}}^{\text{hw}} = \frac{\mathcal{C}}{\mathcal{D}} \times \frac{q_{\text{hw}}}{q_{\text{GPU}}} \times \text{emb}_{\text{imp}}^{\text{hw unit}},$$

where $\mathcal{C}$ is the total development compute in GPU-hours; $q_{\text{hw}}$ is the quantity of component *hw* per node, with *hw* one of: GPU, CPU, RAM, SSD1, SSD2, PSU, motherboard, case, or assembly; and where $\mathcal{D} = \text{lifespan} \times \text{utilization\_rate}$ is the total duration of use of the hardware equipment throughout its lifespan, in hours. We assume an equipment lifespan of four years, in line with values employed in related work (Morand et al., 2024; Schneider et al., 2025; Falk et al., 2025; Desroches et al., 2025), and a reasonable average utilization rate of 0.6 (Luccioni et al., 2023; Wu et al., 2022). The values of $\mathcal{C}$ and $q_{\text{hw}}$ can be found in tab. 6.

## C Sensitivity Analysis

In this section, we subject our environmental assessment to a sensitivity analysis on four parameters, namely the Power Usage Effectiveness (PUE) of the datacenter, the average GPU utilization across runs, the lifespan

---

[6]Only applies to the United States, where the share of *other renewables* is 0.4%, and France, where it is 0.1%.

Table 6: **Environmental assessment variables.** Definitions, values, and sources of the main variables used in the environmental assessment of Moshi. For sources marked with $*$, values are computed instead of taken directly.

| Variable | Notation | Value | Unit | Source |
|---|---|---|---|---|
| Total compute | $\mathcal{C}$ | 3.26e6 | GPU-hours | Kyutai logs |
| Hardware quantity per node | $q_{\mathrm{GPU}}$ | 8 | - | (NVIDIA, 2025a) |
| | $q_{\mathrm{CPU}}$ | 2 | - | |
| | $q_{\mathrm{RAM}}$ | 32 | modules | |
| | $q_{\mathrm{SSD1}}$ | 2 | disks | |
| | $q_{\mathrm{SSD2}}$ | 8 | disks | |
| | $q_{\mathrm{PSU}}$ | 6 | - | |
| | $q_{\mathrm{motherboard}}$ | 1 | - | |
| | $q_{\mathrm{case}}$ | 1 | - | |
| | $q_{\mathrm{assembly}}$ | 1 | - | By definition |
| | $q_{\mathrm{other}}$ | 1 | - | |
| Average GPU utilization | $u_{\mathrm{GPU}}$ | 9.50e-1 | - | Kyutai estimates |
| Average CPU utilization | $u_{\mathrm{CPU}}$ | 5.00e-2 | - | |
| GPU thermal design power | $\mathrm{TDP}_{\mathrm{GPU}}$ | 7.00e2 | W | (TechPowerUp, 2022) |
| CPU thermal design power | $\mathrm{TDP}_{\mathrm{CPU}}$ | 3.50e2 | | (TechPowerUp, 2023) |
| RAM module power | $P_{\mathrm{RAM}}$ | 3.94e0 | | (CodeCarbon, 2026; NVIDIA, 2025a)$*$ |
| Other node hardware power | $P_{\mathrm{other}}$ | 3.77e3 | | (NVIDIA, 2025a)$*$ |
| Power usage effectiveness | PUE | 1.25e0 | - | (Scaleway, 2025) |
| Water usage effectiveness | WUE | 2.50e-1 | L/kWh | (Scaleway, 2025) |
| Cluster overheads | $o_{\mathrm{cluster}}$ | 1.11e0 | - | (NVIDIA, 2025c)$*$ |
| Carbon intensity (2024) | $\mathrm{CI}_{\mathrm{SE}}$ | 3.50e1 | gCO$_2$eq/kWh | (EMBER, 2025a) |
| | $\mathrm{CI}_{\mathrm{FR}}$ | 4.10e1 | | |
| | $\mathrm{CI}_{\mathrm{US}}$ | 3.84e2 | | |
| | $\mathrm{CI}_{\mathrm{AU}}$ | 5.54e2 | | |
| | $\mathrm{CI}_{\mathrm{CN}}$ | 5.55e2 | | |
| | $\mathrm{CI}_{\mathrm{PL}}$ | 6.12e2 | | |
| Electricity water intensity factor (2024) | $\mathrm{EWIF}_{\mathrm{SE}}$ | 4.94e0 | L/kWh | (EMBER, 2025b; Reig et al., 2020)$*$ |
| | $\mathrm{EWIF}_{\mathrm{FR}}$ | 3.34e0 | | |
| | $\mathrm{EWIF}_{\mathrm{US}}$ | 2.30e0 | | |
| | $\mathrm{EWIF}_{\mathrm{AU}}$ | 2.99e0 | | |
| | $\mathrm{EWIF}_{\mathrm{CN}}$ | 4.57e0 | | |
| | $\mathrm{EWIF}_{\mathrm{PL}}$ | 1.12e0 | | |
| Fossil depletion per kWh (FR) | $\mathrm{ADPf}_{\mathrm{kWh}}$ | 9.31e0 | MJ/kWh | (ADEME, 2023) |
| Mineral and metal depletion per kWh (FR) | $\mathrm{ADPe}_{\mathrm{kWh}}$ | 4.86e-8 | kgSbeq/kWh | (ADEME, 2023) |
| Renewable-generated electricity (FR, 2024) | %renewable | 2.72e-1 | - | (EMBER, 2026) |

of the datacenter equipment, and the average utilization of the equipment throughout its lifespan. The two former parameters affect electricity consumption and hence the operational impacts of the project, whereas the two latter parameters affect the fraction of embodied impacts allocated to each hour of use of the equipment, and by extension the embodied impacts of the project.

While the absolute environmental estimates may vary, the qualitative conclusions of this work remain unchanged, in particular the dominance of pre-training and the relatively small contribution of final training to the overall compute budget.

## C.1 Operational Impacts

We first quantify how each environmental indicator varies for three PUE values (1.1, 1.25, and 1.4), corresponding to high-, medium-, and lower-efficiency datacenters, and for GPU utilization rates ranging from 50% to 100%, including our baseline assumption of 95% (tabs. 9 to 13).

Table 7: **Embodied impact factors.** Impact factors provided by Boavizta (Simon et al., 2025) and the ADEME agency (Lees-Perasso et al., 2026) to estimate global warming potential (GWP), primary energy (PE), and abiotic depletion potential (ADP) impacts of hardware production and transport. We abbreviate *Motherboard* as *MoBo*.

| | Boavizta | | | | | | | | | ADEME |
|---|---|---|---|---|---|---|---|---|---|---|
| | **RAM, SSD** die | **RAM** base | **SSD** base | **CPU** die | base | **PSU** | **MoBo** | **Case** | **Assembly** | **GPU** |
| **GWP** kgCO$_2$eq | 2.20e0 per cm$^2$ | 5.22e0 | 6.34e0 | 1.97e0 per cm$^2$ | 9.14e0 | 2.43e1 per kg | 6.61e1 | 1.50e2 | 6.68e0 | 2.70e2 |
| **PE** MJ | 2.73e1 per cm$^2$ | 7.40e1 | 7.40e1 | 2.65e1 per cm$^2$ | 1.56e2 | 3.52e2 per kg | 8.36e2 | 2.20e3 | 6.86e1 | 3.69e3 |
| **ADP** kgSbeq | 6.30e-5 per cm$^2$ | 1.69e-3 | 5.63e-4 | 5.87e-7 per cm$^2$ | 2.04e-2 | 8.30e-3 per kg | 3.69e-3 | 2.02e-2 | 1.41e-6 | 8.94e-3 |

Table 8: **Compute node component specifications.** Hardware specifications of the NVIDIA DGX H100 compute node (NVIDIA, 2025a). We omit fans, network cards, and NVSwitches. For SSD memory density, we select a value from the Boavizta repository reflecting high-end SSDs before the release of the DGX H100.

| Component | Model/Type | Specification | Value | Unit | Source |
|---|---|---|---|---|---|
| CPU | Intel Xeon Platinum 8480C | die size | 19.08 | cm$^2$ | (TechPowerUp, 2023) |
| GPU | NVIDIA H100 SXM HBM3 | die size VRAM capacity VRAM density | 8.14 80 1.65 | cm$^2$ GB GB/cm$^2$ | (TechPowerUp, 2022) (NVIDIA, 2025a) (Moon et al., 2023) |
| RAM | DDR5 | capacity density | 64 2.66 | GB GB/cm$^2$ | (NVIDIA, 2025a) (Choe, 2022) |
| SSD1 | NVMe M.2 | capacity density | 1920 128 | GB GB/cm$^2$ | (NVIDIA, 2025a) (Simon et al., 2025; Choe, 2021) |
| SSD2 | NVMe U.2 | capacity density | 3840 128 | GB GB/cm$^2$ | (NVIDIA, 2025a) (Simon et al., 2025; Choe, 2021) |
| PSU | - | weight | 3 | kg | (Simon et al., 2025) |

**GPU Utilization.** As shown in the tables (middle row, PUE 1.25), even in an extreme scenario where GPUs operate at an average utilization of only 50% across all experiments, our impact estimates decrease by 27% for primary energy and water consumption, and by at most 18% for global warming potential. Under a more realistic average utilization of 70%, these reductions become 15% and 10%, respectively.

**PUE.** Training on datacenters with different energy efficiencies (PUE 1.1 or 1.4) changes our total impact estimates by approximately 8–12%, depending on the indicator. Since abiotic depletion potential is dominated by embodied impacts, its total estimated impact varies by less than 1% across all PUE settings.

Table 9: **Energy consumption sensitivity to PUE and GPU utilization.** Sensitivity analysis on total energy consumption when varying datacenter Power Usage Effectiveness (PUE) and average GPU utilization across runs.

| | Total energy consumption (GWh) | | | | | |
|---|---|---|---|---|---|---|
| | GPU Utilization | | | | | |
| **PUE** | **50%** | **60%** | **70%** | **80%** | **95%** | **100%** |
| **1.1** | 3.4 (-36%) | 3.6 (-31%) | 3.9 (-25%) | 4.2 (-20%) | 4.6 (-12%) | 4.7 (-9%) |
| **1.25** | 3.8 (-27%) | 4.1 (-21%) | 4.4 (-15%) | 4.8 (-9%) | **5.2** | 5.4 (+3%) |
| **1.4** | 4.3 (-18%) | 4.6 (-12%) | 5.0 (-5%) | 5.3 (+2%) | 5.9 (+12%) | 6.0 (+15%) |

Table 10: **Primary energy sensitivity to PUE and GPU utilization.** Sensitivity analysis on total primary energy when varying datacenter Power Usage Effectiveness (PUE) and average GPU utilization across runs.

| | Total primary energy (TJ) | | | | | |
|---|---|---|---|---|---|---|
| | GPU Utilization | | | | | |
| **PUE** | **50%** | **60%** | **70%** | **80%** | **95%** | **100%** |
| **1.1** | 44.3 (-35%) | 47.8 (-30%) | 51.4 (-25%) | 55.0 (-20%) | 60.3 (-12%) | 62.1 (-9%) |
| **1.25** | 50.1 (-27%) | 54.2 (-21%) | 58.2 (-15%) | 62.3 (-9%) | **68.3** | 70.4 (+3%) |
| **1.4** | 56.0 (-18%) | 60.5 (-11%) | 65.0 (-5%) | 69.6 (+2%) | 76.4 (+12%) | 78.6 (+15%) |

Table 11: **Global warming potential sensitivity to PUE and GPU utilization.** Sensitivity analysis on total global warming potential when varying datacenter Power Usage Effectiveness (PUE) and average GPU utilization across runs.

| | Total global warming potential (tCO$_2$eq) | | | | | |
|---|---|---|---|---|---|---|
| | GPU Utilization | | | | | |
| **PUE** | **50%** | **60%** | **70%** | **80%** | **95%** | **100%** |
| **1.1** | 241.9 (-24%) | 253.3 (-21%) | 264.6 (-17%) | 276.0 (-13%) | 293.0 (-8%) | 298.7 (-6%) |
| **1.25** | 260.5 (-18%) | 273.5 (-14%) | 286.4 (-10%) | 299.3 (-6%) | **318.7** | 325.1 (+2%) |
| **1.4** | 279.2 (-12%) | 293.6 (-8%) | 308.1 (-3%) | 322.6 (+1%) | 344.3 (+8%) | 351.5 (+10%) |

## C.2 Embodied Impacts

We additionally analyze the sensitivity of embodied impacts to two key assumptions: hardware lifespan and average hardware utilization over its lifetime. We consider hardware lifespans ranging from three to eight years, and utilization rates between 30% and 100%, the latter corresponding to an idealized upper bound not achievable in practice. In the main text, we have assumed a lifespan of four years (Morand et al., 2024; Schneider et al., 2025; Falk et al., 2025; Desroches et al., 2025) and an utilization of 60% (Luccioni

Table 12: **Abiotic depletion potential sensitivity to PUE and GPU utilization.** Sensitivity analysis on total abiotic depletion potential when varying datacenter Power Usage Effectiveness (PUE) and average GPU utilization across runs.

**Total abiotic depletion potential (kgSbeq)**

| PUE | GPU Utilization | | | | | |
|---|---|---|---|---|---|---|
| | **50%** | **60%** | **70%** | **80%** | **95%** | **100%** |
| **1.1** | 8.1 (-1%) | 8.1 (-1%) | 8.1 (-1%) | 8.2 (-1%) | 8.2 | 8.2 |
| **1.25** | 8.1 (-1%) | 8.2 (-1%) | 8.2 | 8.2 | **8.2** | 8.2 |
| **1.4** | 8.2 | 8.2 | 8.2 | 8.2 | 8.2 | 8.3 (+1%) |

Table 13: **Water consumption sensitivity to PUE and GPU utilization.** Sensitivity analysis on total water consumption when varying datacenter Power Usage Effectiveness (PUE) and average GPU utilization across runs.

**Total water consumption (ML)**

| PUE | GPU Utilization | | | | | |
|---|---|---|---|---|---|---|
| | **50%** | **60%** | **70%** | **80%** | **95%** | **100%** |
| **1.1** | 11.9 (-35%) | 12.9 (-30%) | 13.9 (-25%) | 14.9 (-19%) | 16.4 (-11%) | 16.9 (-9%) |
| **1.25** | 13.5 (-27%) | 14.6 (-21%) | 15.7 (-15%) | 16.8 (-9%) | **18.5** | 19.1 (+3%) |
| **1.4** | 15.0 (-19%) | 16.2 (-12%) | 17.5 (-6%) | 18.7 (+1%) | 20.6 (+11%) | 21.2 (+15%) |

et al., 2023; Wu et al., 2022). Since embodied impacts are allocated proportionally to hardware usage, both parameters affect all embodied indicators in the same way, as shown in tab. 14. We further report how these variations propagate to the total impacts for primary energy, global warming potential, and abiotic depletion potential.

Primary energy (tab. 15) is dominated by operational impacts (fig. 9a), meaning that variations in hardware lifespan and utilization have only a limited influence on the total estimate (less than 3%). In contrast, abiotic depletion potential (tab. 17) is largely dominated by embodied impacts (fig. 9d), and therefore closely follows embodied-impact variations. Global warming potential (tab. 16) lies between these two extremes (fig. 9b). For this reason, we focus the following discussion on global warming potential, which provides the most balanced view of both operational and embodied contributions.

**Equipment Lifespan.** As shown in tab. 14, 60% utilization column, a hardware lifespan of six years, two more than our assumption, would reduce our embodied impact estimates by 33%. Conversely, reducing the lifespan by one year would increase all embodied impacts by 33%. For global warming potential impacts (tab. 16), increasing the lifespan to six years reduces our estimate by 11%, and reducing the lifespan to three years increases our estimate also by 11%.

**Equipment Utilization during Lifespan.** Tab. 14 (four years row) shows that our embodied impact estimates would be 50% higher if the hardware was only used for 40% of its lifespan before retirement, instead of our assumed 60%. In an extreme case where the hardware was only leveraged for 30% of its lifespan, our estimates would be doubled. On the other hand, increasing the utilization rate to 90%, close to the theoretical maximum, would reduce our embodied impact estimates by one-third. Total global warming potential impacts (tab. 16) would increase by 17% for 40% utilization and by 33% for 30% utilization, and would decrease only by 11% for 90% utilization.

**Commentary.** Embodied impact estimates are more sensitive to parameter assumptions (inverse scaling with hardware lifespan and utilization rate), whereas operational impact estimates show a lower variation upon changing assumed GPU utilization and datacenter PUE (linear scaling).

Table 14: **Embodied impact sensitivity to equipment lifespan and utilization.** Relative change in embodied impacts when varying the assumed lifespan of the hardware equipment, and the average utilization of the equipment during its lifespan. Same for all environmental impact indicators.

**Change in embodied impacts (same for all indicators)**

| Lifespan | Equipment utilization during lifespan | | | | | | | |
|---|---|---|---|---|---|---|---|---|
| | 30% | 40% | 50% | 60% | 70% | 80% | 90% | 100% |
| **3 years** | +167% | +100% | +60% | +33% | 14% | 0% | -11% | -20% |
| **4 years** | +100% | +50% | +20% | − | -14% | -25% | -33% | -40% |
| **5 years** | +60% | +20% | -4% | -20% | -31% | -40% | -47% | -52% |
| **6 years** | +33% | 0% | -20% | -33% | -43% | -50% | -56% | -60% |
| **7 years** | +14% | -14% | -31% | -43% | -51% | -57% | -62% | -66% |
| **8 years** | 0% | -25% | -40% | -50% | -57% | -63% | -67% | -70% |

Table 15: **Primary energy sensitivity to equipment lifespan and utilization.** Sensitivity analysis on total primary energy when varying the assumed lifespan of the hardware equipment, and the average utilization of the equipment during its lifespan.

**Total primary energy (TJ)**

| Lifespan | Equipment utilization during lifespan | | | | | | | |
|---|---|---|---|---|---|---|---|---|
| | 30% | 40% | 50% | 60% | 70% | 80% | 90% | 100% |
| **3 years** | 70.7 (3%) | 69.7 (2%) | 69.2 (1%) | 68.8 (1%) | 68.5 | 68.3 | 68.2 | 68.1 |
| **4 years** | 69.7 (2%) | 69.0 (1%) | 68.6 | **68.3** | 68.1 | 68.0 (-1%) | 67.9 (-1%) | 67.8 (-1%) |
| **5 years** | 69.2 (1%) | 68.6 | 68.3 | 68.1 | 67.9 (-1%) | 67.8 (-1%) | 67.7 (-1%) | 67.6 (-1%) |
| **6 years** | 68.8 (1%) | 68.3 | 68.1 | 67.9 (-1%) | 67.8 (-1%) | 67.7 (-1%) | 67.6 (-1%) | 67.5 (-1%) |
| **7 years** | 68.5 | 68.1 | 67.9 (-1%) | 67.8 (-1%) | 67.6 (-1%) | 67.6 (-1%) | 67.5 (-1%) | 67.4 (-1%) |
| **8 years** | 68.3 | 68.0 (-1%) | 67.8 (-1%) | 67.7 (-1%) | 67.6 (-1%) | 67.5 (-1%) | 67.4 (-1%) | 67.4 (-1%) |

Table 16: **Global warming potential sensitivity to equipment lifespan and utilization.** Sensitivity analysis on global warming potential when varying the assumed lifespan of the hardware equipment, and the average utilization of the equipment during its lifespan.

**Total global warming potential (tCO$_2$eq)**

| Lifespan | Equipment utilization during lifespan | | | | | | | |
|---|---|---|---|---|---|---|---|---|
| | 30% | 40% | 50% | 60% | 70% | 80% | 90% | 100% |
| **3 years** | 494.0 (+55%) | 423.9 (+33%) | 381.8 (+20%) | 353.7 (+11%) | 333.7 (+5%) | 318.7 | 307.0 (-4%) | 297.6 (-7%) |
| **4 years** | 423.9 (+33%) | 371.3 (+17%) | 339.7 (+7%) | **318.7** | 303.6 (-5%) | 292.4 (-8%) | 283.6 (-11%) | 276.6 (-13%) |
| **5 years** | 381.8 (+20%) | 339.7 (+7%) | 314.5 (-1%) | 297.6 (-7%) | 285.6 (-10%) | 276.6 (-13%) | 269.6 (-15%) | 264.0 (-17%) |
| **6 years** | 353.7 (+11%) | 318.7 | 297.6 (-7%) | 283.6 (-11%) | 273.6 (-14%) | 266.1 (-17%) | 260.2 (-18%) | 255.6 (-20%) |
| **7 years** | 333.7 (+5%) | 303.6 (-5%) | 285.6 (-10%) | 273.6 (-14%) | 265.0 (-17%) | 258.6 (-19%) | 253.5 (-20%) | 249.5 (-22%) |
| **8 years** | 318.7 | 292.4 (-8%) | 276.6 (-13%) | 266.1 (-17%) | 258.6 (-19%) | 252.9 (-21%) | 248.5 (-22%) | 245.0 (-23%) |

Table 17: **Abiotic depletion potential sensitivity to equipment lifespan and utilization.** Sensitivity analysis on total abiotic depletion potential when varying the assumed lifespan of the hardware equipment, and the average utilization of the equipment during its lifespan.

**Total abiotic depletion potential (kgSbeq)**

| Lifespan | Equipment utilization during lifespan | | | | | | | |
|---|---|---|---|---|---|---|---|---|
| | 30% | 40% | 50% | 60% | 70% | 80% | 90% | 100% |
| **3 years** | 21.5 (+162%) | 16.2 (+97%) | 13.0 (+58%) | 10.9 (+32%) | 9.3 (+14%) | 8.2 | 7.3 (-11%) | 6.6 (-19%) |
| **4 years** | 16.2 (+97%) | 12.2 (+49%) | 9.8 (+19%) | **8.2** | 7.1 (-14%) | 6.2 (-24%) | 5.6 (-32%) | 5.0 (-39%) |
| **5 years** | 13.0 (+58%) | 9.8 (+19%) | 7.9 (-4%) | 6.6 (-19%) | 5.7 (-30%) | 5.0 (-39%) | 4.5 (-45%) | 4.1 (-50%) |
| **6 years** | 10.9 (+32%) | 8.2 | 6.6 (-19%) | 5.6 (-32%) | 4.8 (-41%) | 4.2 (-48%) | 3.8 (-54%) | 3.4 (-58%) |
| **7 years** | 9.3 (+14%) | 7.1 (-14%) | 5.7 (-30%) | 4.8 (-41%) | 4.2 (-49%) | 3.7 (-55%) | 3.3 (-60%) | 3.0 (-64%) |
| **8 years** | 8.2 | 6.2 (-24%) | 5.0 (-39%) | 4.2 (-48%) | 3.7 (-55%) | 3.2 (-61%) | 2.9 (-65%) | 2.6 (-68%) |

