# OpenReview forum: "Environmental Footprint of GenAI Research: Insights from the Moshi Foundation Model"
_TMLR — Decision pending for TMLR_

### Review · Reviewer_UMSy · 2026-05-29

**Summary Of Contributions:**

This paper is the first (to my knowledge) comprehensive resource consumption/environmental impact analysis for a strong multimodal model, based on real-world training and development logs and data. They provide detailed analyses across all stages of creating these multimodal models, including the LM backbone, data generation, failed/broken runs, and (the bulk of the compute) training the final model. They look at all training stages like pretraining, post-training, and finetuning, across different stages of the R&D process (debugging, development, final training runs, ablations, evaluations), for all parts of their model, and they then perform a multi-part lifecycle assessment to understand the full environmental impact based on their measurements.

Strengths include:
* The report is very comprehensive, at a level of granularity that is rarely (if ever) released publicly.
* They perform a *multi-impact* lifecycle assessment, analyzing more than just the primary power draw of the equipment.
* The model analyzed is of a type that has never really been evaluated at this level of granularity.

Weaknesses:
* They seem to assume that all GPUs are operating at 95% power draw, which is generally quite high compared to real-world measurements in other works.
* No estimates of embodied water consumption

**Audience:**

Yes

**Audience Explanation:**

Clearly yes. The environmental cost of AI is already of high interest to the overall community, and it is growing quickly still. Their R&D breakdown of a MLLM is new, and relevant to ML systems, sustainability, policy, and multimodal researchers/readers.

**Broader Impact Concerns:**

No separate statement is required; the work in general is a positive to the community.

**Claims And Evidence:**

Yes

**Claims Explanation:**

Overall yes. The paper's central claims come from direct measurements of GPU hours spent from original/internal logs of the model's development process. The distribution of compute described in Section 3 come directly from these logs (which will ideally be released publicly after publication?), which clearly corroborate the paper's claims (e.g. under 4% of compute was spent on the final training run(s), 13% of the runs taking 89% of the compute, etc).

My main concern is in the environmental assessment in Section 4, which relies on estimated GPU power consumption rather than measured data. If I understand correctly, the authors assume that each GPU, when active, draws 95% of its rated TDP at all times, with the 5% reduction meant to cover idle periods. In practice, H100 power draw varies substantially across workloads, and exploratory work can sometimes be much less compute-efficient (i.e. drawing a low % of TDP due to under-optimized code), and this could have an especially large impact on the estimates in this work given the large percentage of GPU hours dedicated to runs besides the final training runs. In other words, the power usage and most of the environmental impact assessments are probably over-estimates to some degree. To be clear: the GPU hour breakdown per stage is not affected by this at all, and that alone is a clearly interesting reported result. My only uncertainty/reservation is with the power usage estimates.

To address this, I would recommend the authors, if possible:
* Rerun a few small-scale experiments mirroring different stages, measuring direct power draw (using code carbon, as they mentioned, or wandb system metrics, etc) to get estimates for how power-intensive different stages are.
* If that is not possible, I think reporting a range would also be quite helpful, rather than assuming a flat 95%.

**Requested Changes:**

Critical:
* Make the GPU power assumption more explicit in the main text, and address the likely bias (i.e. overestimate) from it. Because real H100 power draw varies with the workload, and because under-optimized workloads generally draw far below the max power draw of an H100, I think it's worth also adding a sensitivity analysis that would show what would happen with a much lower power draw (e.g. 60 or 70% of the max TDP). If any direct measurements are available (or could be obtained), these would also strengthen the overall analysis.
* It's worth stating more clearly what the definition of "final" and "development" compute in the work is, as the 4% figure for final runs reported is much lower than that for BLOOM and Olmo 1/2. The explanations offered (e.g. building these systems from scratch, novelty of the model, etc) are reasonable and probably accurate, but a strong and up front definition of the boundary here would make the claims even stronger.

Would strengthen:
* Mentioning uncertainty in other factors would also be worthwhile, including hardware lifespan, or PUE. On the subject of PUE, 1.25 seems a bit high for modern AI data centers.
* Add a comment about omitting an estimate for embodied water emissions; semiconductor manufacturing is generally quite water-intensive, though the data on it is quite opaque.
* There has been a bit of follow up work that is worth mentioning/engaging with; e.g. a follow up report covering Olmo 3 (including post-training) was recently released: https://arxiv.org/abs/2605.01158 and concurrent work from Epoch.AI found similar trends as this work, showing quite high (>80%) fractions of compute dedicated to R&D before the final training runs.
* Minor correction: The introduction classifies Moshi as a speech-to-text model, but later on it's described as a speech-to-speech model.

---

> ### Author Response · Authors · 2026-06-16
> **Reply to reviewer UMSy**
>
> **Data release.** We have obtained authorization from Kyutai to release an anonymized version of the training logs: https://anonymous.4open.science/r/GenAIFootprint-0C9C. The released data include, for each run, the number of requested GPUs; the training, validation, evaluation, and generation durations; the creation and modification times of the log entry; a training phase identifier; and tags for run classification (debug, final, ablation, etc.). We compliment the data file with metadata explaining the meaning of each field and how it was derived, if applicable.
>
> Example of a log entry:
>
> ```
> {
>     "id": 57,
>     "gpus": 2,
>     "durations": {
>         "train": 2.9230795332,
>         "valid": 0.2250824352,
>         "evaluate": 0.6980337475,
>         "generate": 0.0022963946
>     },
>     "ctime": 1702994663,
>     "mtime": 1702883370,
>     "phase": "A",
>     "tags": "debug"
> }
> ```
>
> **Note:** while preparing the data for release, we refined the preprocessing of the raw logs, in particular by improving the removal of duplicate experiment entries. This led to the reclassification of a small number of experiments, resulting in minor updates to Figure 4. The remaining figures and conclusions are essentially unchanged, and the environmental assessment is unaffected by these modifications.
>
> **GPU power draw.** We agree with the reviewer that assuming a constant GPU utilization close to peak TDP likely leads to an overestimate of operational impacts for some experiment types, particularly exploratory or under-optimized runs. We have therefore clarified this assumption and its limitations in Section 4.1 as follows:
>
> **Before**
> > GPU energy consumption is estimated as the product of the number of concurrently used GPUs, the maximum rated GPU power, and a 95% utilization factor, which accounts for brief periods of non-GPU-intensive work.
>
> **After**
> > We first estimate GPU electricity consumption as the product of compute (GPU-time, defined as run duration multiplied by the number of GPUs used, aggregated across all runs), the maximum rated GPU power, and a GPU utilization factor. We set this utilization factor to 95%, based on observations from Kyutai indicating near-full GPU utilization during most runs, while accounting for brief periods of non-GPU-intensive work. However, these observations may not hold uniformly across all run types, particularly during under-optimized early experimentation or debugging runs. As a result, assuming a constant utilization factor close to full utilization for all runs may overestimate electricity consumption. We therefore provide a sensitivity analysis of this parameter in sec. C.
>
> Unfortunately, retrospective power measurements are not possible for the experiments considered in this study. Following the reviewer’s suggestion, we therefore added a sensitivity analysis varying GPU utilization, datacenter PUE, and hardware lifespan/utilization assumptions.
>
> More specifically, we quantify how each environmental indicator varies for three PUE values (1.1, 1.25, and 1.4), corresponding to high-, medium-, and lower-efficiency datacenters, and for GPU utilization rates ranging from 50% to 100%, including our baseline assumption of 95% (Tables A-E). While the absolute environmental estimates vary accordingly, the qualitative conclusions of the paper remain unchanged, in particular the dominance of pre-training and the relatively small contribution of final training to the overall compute budget.
>
> *GPU utilization.* As shown in the tables below (middle row, PUE 1.25), even in an extreme scenario where GPUs operate at an average utilization of only 50% across all experiments, our impact estimates decrease by 27% for primary energy and water consumption, and by at most 18% for global warming potential. Under a more realistic average utilization of 70%, these reductions become 15% and 10%, respectively.
>
> *PUE.* Training on datacenters with different energy efficiencies (PUE 1.1 or 1.4) changes our total impact estimates by approximately 8-12%, depending on the indicator. Since abiotic depletion potential is dominated by embodied impacts, its total estimated impact varies by less than 1% across all PUE settings.

---

> > ### Author Response · Authors · 2026-06-16
> > **Reply to reviewer UMSy - 2**
> >
> > **Table A: Total energy consumption (GWh) when varying PUE and GPU usage**
> >
> > | PUE\GPU utilization | 50%         | 60%         | 70%         | 80%         | *95%*       | 100%        |
> > | ------------- | ----------- | ----------- | ----------- | ----------- | ----------- | ----------- |
> > | **1.1**       | 3.4 (-36%) | 3.6 (-31%) | 3.9 (-25%) | 4.2 (-20%) | 4.6 (-12%) | 4.7 (-9%)  |
> > | ***1.25***    | 3.8 (-27%) | 4.1 (-21%) | 4.4 (-15%) | 4.8 (-9%)  |  ***5.2***  | 5.4 (+3%)  |
> > | **1.4**       | 4.3 (-18%) | 4.6 (-12%) | 5.0 (-5%)  | 5.3 (+2%)  | 5.9 (+12%) | 6.0 (+15%) |
> >
> > **Table B: Total primary energy (TJ) when varying PUE and GPU usage**
> >
> > | PUE\GPU utilization | 50%          | 60%         | 70%         | 80%         |  *95%*       | 100%        |
> > | ------------- | ------------ | ----------- | ----------- | ----------- | ----------- | ----------- |
> > | **1.1**       | 44.3 (-35%) | 47.8 (-30%) | 51.4 (-25%) | 55.0 (-20%) | 60.3 (-12%) | 62.1 (-9%)  |
> > |  ***1.25***    | 50.1 (-27%) | 54.2 (-21%) | 58.2 (-15%) | 62.3 (-9%)  |  ***68.3***  | 70.4 (+3%)  |
> > | **1.4**       | 56.0 (-18%) | 60.5 (-11%) | 65.0 (-5%)  | 69.6 (+2%)  | 76.4 (+12%) | 78.6 (+15%) |
> >
> > **Table C: Total global warming potential (tCO2eq) when varying PUE and GPU usage**
> >
> > | PUE\GPU utilization | 50%           | 60%          | 70%          | 80%          |  *95%*       | 100%         |
> > | ------------- | ------------- | ------------ | ------------ | ------------ | ----------- | ------------ |
> > | **1.1**       | 241.9 (-24%) | 253.3 (-21%) | 264.6 (-17%) | 276.0 (-13%) | 293.0 (-8%) | 298.7 (-6%)  |
> > |  ***1.25***    | 260.5 (-18%) | 273.5 (-14%) | 286.4 (-10%) | 299.3 (-6%)  |  ***318.7*** | 325.1 (+2%)  |
> > | **1.4**       | 279.2 (-12%) | 293.6 (-8%)  | 308.1 (-3%)  | 322.6 (+1%)  | 344.3 (+8%) | 351.5 (+10%) |
> >
> >
> > **Table D: Total water consumption (ML) when varying PUE and GPU usage**
> >
> > | PUE\GPU utilization | 50%         | 60%         | 70%         | 80%         |  *95%*       | 100%        |
> > | ------------- | ----------- | ----------- | ----------- | ----------- | ----------- | ----------- |
> > | **1.1**       | 11.9 (-35%) | 12.9 (-30%) | 13.9 (-25%) | 14.9 (-19%) | 16.4 (-11%) | 16.9 (-9%)  |
> > |  ***1.25***    | 13.5 (-27%) | 14.6 (-21%) | 15.7 (-15%) | 16.8 (-9%)  |  ***18.5***  | 19.1 (+3%)  |
> > | **1.4**       | 15.0 (-19%) | 16.2 (-12%) | 17.5 (-6%)  | 18.7 (+1%)  | 20.6 (+11%) | 21.2 (+15%) |
> >
> >
> > **Table E: Total abiotic depletion potential (kgSbeq) when varying PUE and GPU usage**
> >
> > | PUE\GPU utilization | 50%        | 60%        | 70%        | 80%        |  *95%*      | 100%       |
> > | ------------- | ---------- | ---------- | ---------- | ---------- | ---------- | ---------- |
> > | **1.1**       | 8.1 (-1%) | 8.1 (-1%) | 8.1 (-1%) | 8.2 (-1%) | 8.2       | 8.2       |
> > |  ***1.25***    | 8.1 (-1%) | 8.2 (-1%) | 8.2       | 8.2       |  ***8.2*** | 8.22       |
> > | **1.4**       | 8.2       | 8.2       | 8.2       | 8.2       | 8.2       | 8.3 (+1%) |
> >
> > *Impact of Hardware Lifespan and Utilization.* We additionally analyze the sensitivity of embodied impacts to two key assumptions: hardware lifespan and average hardware utilization over its lifetime. We consider hardware lifespans ranging from three to eight years, and utilization rates between 30% and 100%, the latter corresponding to an idealized upper bound not achievable in practice. In the paper, we have assumed a lifespan of ***four years*** (Morand et al., 2024; Schneider et al., 2025; Falk et al., 2025; Desroches et al., 2025) and an utilization of ***60%*** (Luccioni et al., 2023; Wu et al., 2022). Since embodied impacts are allocated proportionally to hardware usage, both parameters affect all embodied indicators in the same way, as shown in Table F. We further report how these variations propagate to the total impacts for primary energy, global warming potential, and abiotic depletion potential.
> >
> > Primary energy (Table H) is dominated by operational impacts (Fig. 9a), meaning that variations in hardware lifespan and utilization have only a limited influence on the total estimate (less than 3%). In contrast, abiotic depletion potential (Table G) is largely dominated by embodied impacts (Fig. 9d), and therefore closely follows embodied-impact variations. Global warming potential (Table I) lies between these two extremes (Fig. 9b). For this reason, we focus the following discussion on global warming potential, which provides the most balanced view of both operational and embodied contributions.
> >
> > *Lifespan.* As shown in Table F, 60% utilization column, a hardware lifespan of six years, two more than our assumption, would reduce our embodied impact estimates by 33%. Conversely, reducing the lifespan by one year would increase all embodied impacts by 33%. For global warming potential impacts (Table I), increasing the lifespan to six years reduces our estimate by 11%, and reducing the lifespan to three years increases our estimate also by 11%.

---

> > > ### Author Response · Authors · 2026-06-16
> > > **Reply to reviewer UMSy - 3**
> > >
> > > *Utilization.* Table F (four years row) shows that our embodied impact estimates would be 50% higher if the hardware was only used for 40% of its lifespan before retirement, instead of our assumed 60%. In an extreme case where the hardware was only leveraged for 30% of its lifespan, our estimates would be doubled. On the other hand, increasing the utilization rate to 90%, close to the theoretical maximum, would reduce our embodied impact estimates by one-third. Total global warming potential impacts (Table I) would increase by 17% for 40% utilization and by 33% for 30% utilization, and would decrease only by 11% for 90% utilization.
> > >
> > > *Commentary.* Embodied impact estimates are more sensitive to parameter assumptions (inverse scaling with hardware lifespan and utilization rate), whereas operational impact estimates show a lower variation upon changing assumed GPU utilization and datacenter PUE (linear scaling).
> > >
> > > **Table F: Change in embodied impacts (same for all indicators) when varying hardware lifespan and utilization**
> > >
> > > | Lifespan\Utilization | 30%   | 40%   | 50%  | *60%*  | 70%  | 80%  | 90%  | 100% |
> > > | -------------------- | ----- | ----- | ---- | ------ | ---- | ---- | ---- | ---- |
> > > | **3 years**  | +167% | +100% | +60% | +33%   | 14%  | 0%   | -11% | -20% |
> > > | ***4 years***  | +100% | +50%  | +20% | **-**| -14% | -25% | -33% | -40% |
> > > | **5 years** | +60%  | +20%  | -4%  | -20%   | -31% | -40% | -47% | -52% |
> > > | **6 years** | +33%  | 0%    | -20% | -33%   | -43% | -50% | -56% | -60% |
> > > | **7 years** | +14%  | -14%  | -31% | -43%   | -51% | -57% | -62% | -66% |
> > > | **8 years**  | 0%    | -25%  | -40% | -50%   | -57% | -63% | -67% | -70% |
> > >
> > > **Table G: Change in total abiotic depletion potential when varying hardware lifespan and utilization**
> > >
> > > | Lifespan\Utilization | 30%  | 40% | 50%| *60%* | 70% | 80%| 90%| 100%|
> > > | ------- | ----- | --- | ---- | --- | -- | --- | - | -- |
> > > | **3 years**| 21.5 (+162%) | 16.2 (+97%) | 13.0 (+58%) | 10.9 (+32%) | 9.3 (+14%) | 8.2        | 7.3 (-11%) | 6.6 (-19%) |
> > > | ***4 years***| 16.2 (+97%)  | 12.2 (+49%) | 9.8 (+19%)  | ***8.2***   | 7.1 (-14%) | 6.2 (-24%) | 5.6 (-32%) | 5.0 (-39%) |
> > > | **5 years**| 13.0 (+58%)  | 9.8 (+19%)  | 7.9 (-4%)   | 6.6 (-19%)  | 5.7 (-30%) | 5.0 (-39%) | 4.5 (-45%) | 4.1 (-50%) |
> > > | **6 years**| 10.9 (+32%)  | 8.2         | 6.6 (-19%)  | 5.6 (-32%)  | 4.8 (-41%) | 4.2 (-48%) | 3.8 (-54%) | 3.4 (-58%) |
> > > | **7 years**| 9.3 (+14%)   | 7.1 (-14%)  | 5.7 (-30%)  | 4.8 (-41%)  | 4.2 (-49%) | 3.7 (-55%) | 3.3 (-60%) | 3.0 (-64%) |
> > > | **8 years**| 8.2          | 6.2 (-24%)  | 5.0 (-39%)  | 4.2 (-48%)  | 3.7 (-55%) | 3.2 (-61%) | 2.9 (-65%) | 2.6 (-68%) |
> > >
> > > **Table H: Change in total primary energy when varying hardware lifespan and utilization**
> > >
> > > | Lifespan\Utilization | 30%| 40%| 50%| *60%*| 70%| 80%| 90%| 100%|
> > > | ----- | --- | ---- | --- | ----- | ----------- | ----------- | ----------- | ----------- |
> > > | **3 years**           | 70.7 (3%) | 69.7 (2%)  | 69.2 (1%)  | 68.8 (1%)  | 68.5       | 68.3       | 68.2       | 68.1       |
> > > | ***4 years***        | 69.7 (2%) | 69.0 (1%)  | 68.6       | ***68.3*** | 68.1       | 68.0 (-1%) | 67.9 (-1%) | 67.8 (-1%) |
> > > | **5 years**          | 69.2 (1%) | 68.6       | 68.3       | 68.1       | 67.9 (-1%) | 67.8 (-1%) | 67.7 (-1%) | 67.6 (-1%) |
> > > | **6 years**          | 68.8 (1%) | 68.3       | 68.1       | 67.9 (-1%) | 67.8 (-1%) | 67.7 (-1%) | 67.6 (-1%) | 67.5 (-1%) |
> > > | **7 years**                | 68.5      | 68.1       | 67.9 (-1%) | 67.8 (-1%) | 67.6 (-1%) | 67.6 (-1%) | 67.5 (-1%) | 67.4 (-1%) |
> > > | **8 years**                | 68.3      | 68.0 (-1%) | 67.8 (-1%) | 67.7 (-1%) | 67.6 (-1%) | 67.5 (-1%) | 67.4 (-1%) | 67.4 (-1%) |
> > >
> > > **Table I: Change in total global warming potential when varying hardware lifespan and utilization**
> > >
> > > | Lifespan\Utilization | 30%           | 40%           | 50%           | *60%*         | 70%           | 80%           | 90%           | 100%          |
> > > | -------------------- | ------------- | ------------- | ------------- | ------------- | ------------- | ------------- | ------------- | ------------- |
> > > | **3 years**                | 494.0 (+55%) | 423.9 (+33%) | 381.8 (+20%) | 353.7 (+11%) | 333.7 (+5%)  | 318.7        | 307.0 (-4%)  | 297.6 (-7%)  |
> > > | ***4 years***              | 423.9 (+33%) | 371.3 (+17%) | 339.7 (+7%)  | ***318.7***  | 303.6 (-5%)  | 292.4 (-8%)  | 283.6 (-11%) | 276.6 (-13%) |
> > > | **5 years**                | 381.8 (+20%) | 339.7 (+7%)  | 314.5 (-1%)  | 297.6 (-7%)  | 285.6 (-10%) | 276.6 (-13%) | 269.6 (-15%) | 264.0 (-17%) |
> > > | **6 years**                | 353.7 (+11%) | 318.7        | 297.6 (-7%)  | 283.6 (-11%) | 273.6 (-14%) | 266.1 (-17%) | 260.2 (-18%) | 255.6 (-20%) |
> > > | **7 years**                | 333.7 (+5%)  | 303.6 (-5%)  | 285.6 (-10%) | 273.6 (-14%) | 265.0 (-17%) | 258.6 (-19%) | 253.5 (-20%) | 249.5 (-22%) |
> > > | **8 years**  | 318.7        | 292.4 (-8%)  | 276.6 (-13%) | 266.1 (-17%) | 258.6 (-19%) | 252.9 (-21%) | 248.5 (-22%) | 245.0 (-23%) |

---

> > > > ### Author Response · Authors · 2026-06-16
> > > > **Reply to reviewer UMSy - 4**
> > > >
> > > > **Definitions of final and development.** We have added a reminder of the definitions of final and development compute in the discussion. We have also renamed the paragraph to “Final vs. Total Development Costs”:
> > > > > **Final vs. Total Development Costs:** The *final compute* of Moshi aggregates the GPU-time required to train, in successive phases, the released versions of each of its modules, whereas *total compute* represents the total compute of the project from beginning to end, including the final compute. [...]
> > > >
> > > > **Uncertainty in assumed factors.** We address this in our answer above (**GPU power draw**). As for the PUE of 1.25, that is the value published by Scaleway in its 2024 environmental report for the datacenter where Moshi was developed. It is in fact one of the lowest PUEs among all their datacenters.
> > > >
> > > > **Embodied water emissions.** We do mention the omission in the scope definition at the beginning of Section 4, and in the analysis in Section 4.2 ("We do not estimate embodied water consumption, but it should be considerable due to the ultrapure water and the electricity required for semiconductor manufacturing (Boyd, 2012; Li et al., 2025a)."), but we have modified and extended our justification to emphasize the importance of embodied water consumption for semiconductor manufacturing:
> > > >
> > > > **Before**
> > > > > We further restrict water consumption estimates to the use phase only. Although studies on the water consumption of hardware manufacturing exist (Falk et al., 2025; Boyd, 2012), extrapolating these results to individual hardware components would require additional assumptions and is therefore outside the scope of this work.
> > > >
> > > > **After**
> > > > > We further restrict water consumption estimates to the use phase only. Although studies on the water consumption of hardware manufacturing exist (Falk et al., 2025; Boyd, 2012), extrapolating these results to individual hardware components would require **a level of expertise that is beyond the abilities of the authors. Nevertheless, it should be noted that, as shown in these and related works (Hess, 2024), the water footprint of semiconductor manufacturing is considerable.**
> > > >
> > > > *Additional reference: [Julia Christina Hess. Chip production’s ecological footprint: Mapping climate and environmental impact. Technical report, Interface (Stiftung Neue Verantwortung), 2024.](https://www.interface-eu.org/publications/chip-productions-ecological-footprint)*
> > > >
> > > > **Engagement with follow-up work.** We did see the post-training report of OLMo 3 when it came out; Allen AI is really setting a good example regarding AI impact assessment. The Epoch.AI article is an interesting read, but its focus on monetary cost sets it beyond the scope of our study.
> > > >
> > > > We have cited the OLMo 3 report in Section 2.2 and added a comparison in our discussion as follows:
> > > >
> > > > > **AI System Research and Development.** A few published studies assess the impacts of training suites of LLMs while also considering development overheads. Among these, some report the impact of development activities as a single number (Lakim et al., 2022), whereas others provide breakdowns by model size (Morrison et al., 2025) **and training stage (Morrison et al, 2026)**.
> > > >
> > > > > **Final vs. Total Development Costs.** [...] Morrison et al. (2025) report that training the open source versions of the OLMo suite of LLMs accounted for 50% of the environmental impacts of the project. **A more recent report on the latest generation of OLMo models (Morrison et al., 2026) places final run costs at 18% of the total compute of the project, with individual training phases reaching figures as low as 2%.** [...]
> > > >
> > > > *Additional reference: [Jacob Morrison, Noah A. Smith, and Emma Strubell. The hidden cost of thinking: Energy use and environmental impact of LMs beyond pretraining, 2026.](https://arxiv.org/abs/2605.01158)*
> > > >
> > > > **Moshi modality definition.** Thank you for pointing this mistake out, Moshi is indeed a speech-to-speech model. Note that as Moshi was trained with a combination of text and speech tokens, it is referred to as a multimodal model (speech/text) in Kyutai’s paper despite not accepting diverse modalities as input.

---

### Review · Reviewer_K823 · 2026-06-07

**Summary Of Contributions:**

This paper addresses the timely and important problem by having access to first-hand data for a foundation model. However, the paper's motivation, methodological novelty, and the insights derived from the analysis remain unclear, making it difficult to assess the broader contribution of the work.

**Audience:**

Yes

**Audience Explanation:**

This paper investigates the environmental footprint of a multimodal foundation model and leverages detailed operational data spanning both training and deployment.

**Claims And Evidence:**

No

**Claims Explanation:**

See Requested Changes.

**Requested Changes:**

This paper investigates the environmental footprint of a multimodal foundation model and leverages detailed operational data spanning both training and deployment. While this is a valuable dataset, several aspects of the study require further clarification.

- The motivation for focusing on multimodal large language models is unclear. Prior work has already examined the environmental impacts of conventional LLMs. There is a lack of explanation why multimodal models are expected to differ and what new environmental challenges or characteristics motivate a dedicated study.
- A significant portion of the analysis focuses on computational cost. However, the connection between compute metrics and environmental impact is not clearly established. The paper would benefit from a clearer explanation of how the reported compute analyses contribute to understanding environmental outcomes.
- Functional units are a critical component of lifecycle assessment studies. But, the functional unit used in this work is not clearly defined. What specific service or functionality provided by the multimodal model serves as the basis for normalization? Without a clear functional unit, it is difficult to interpret and compare the reported results.
- The methodological novelty of the work is unclear. It appears that existing environmental modeling techniques are applied to a new system.
- The paper does not clearly specify the lifecycle assessment framework used in the study. Given the existence of multiple LCA methodologies and databases, the exact framework, assumptions, and modeling choices should be described in greater detail.
- Different LCA frameworks support different impact categories and indicators. The rationale for selecting only the four impact indicators reported in Table 2 is not justified. The authors should explain why these indicators were chosen and whether other relevant environmental impact categories were considered.

Overall, while the dataset and problem are valuable, the paper would be strengthened by a clearer motivation, more detailed methodological description, and stronger discussion of the novelty and implications of the findings.

---

> ### Author Response · Authors · 2026-06-16
> **Reply to reviewer K823**
>
> We thank the reviewer for acknowledging the value of our work and for their helpful comments and insights, which allowed us to improve the rigor of the paper. We address each concern below.
>
> **Motivation for studying MLLMs.**  We will clarify that the goal of this paper is not to assess specifically the environmental impacts of MLLMs, nor to argue that they fundamentally differ from conventional LLMs. Rather, our objective is to analyze the often-overlooked computational and environmental costs of the full research and development cycle of modern AI systems, and to pave the way for more exhaustive environmental reporting in AI research.
>
> This study was made possible by unprecedented access to the complete internal training logs of Moshi, a frontier speech-to-speech foundation model developed by Kyutai. In most prior work, such detailed development data is unavailable or not leveraged, meaning that experimentation, debugging, hyperparameter search, architectural exploration, ablation studies, and discarded runs are typically omitted or only briefly discussed (Section 2.2). Moshi therefore provides a rare opportunity to quantify the environmental impacts of all stages of AI model development. We additionally plan to release the processed data to encourage further analyses and promote greater transparency in environmental reporting.
>
> **Link between compute and environmental impacts.** To make the link between compute and environmental impacts clearer, we have extended our Section 4 introduction as follows:
>
> **Before**
> > In this section, we quantify the environmental impact of developing the Moshi model from the first experiments up to the last training run.
>
> **After**
> > In this section, we quantify the environmental impacts of developing the Moshi model, from initial experimentation up to the **final ablations. We estimate both operational impacts, arising from the electricity consumed during training, and embodied impacts, arising from the manufacturing of the hardware where the training was run, based on the total compute of the project. Sec. 4.1 details how compute is converted into environmental impacts.**
> >
> > **More generally, embodied hardware impacts are directly proportional to compute (GPU-time), since they are allocated according to hardware use time. We assume the use time of hardware components other than GPUs to be proportional to GPU-time, as GPUs cannot operate independently from the rest of the compute node.**
> >
> > **Operational impacts are less directly tied to computation, as they depend on the actual electricity consumption of each run. Since such measurements are unavailable retrospectively, we assume a constant average power consumption across runs using the utilization estimates provided by Kyutai’s researchers. In practice, this means that the environmental impact distributions may not exactly match the compute distributions reported in sec. 3, although prior work suggests that the discrepancy remains limited (Morrison et al., 2025).**
>
>
> And we have reformulated Section 4.1 as follows:
>
> **Before**
> > [...] We distinguish between operational impacts, which arise from the use phase of the hardware during model training (i.e., electricity consumption while compute nodes are operating), and embodied impacts, which correspond to impacts incurred during hardware production, transport, and end of life.
> >
> > **Operational Impacts.** GPU energy consumption is estimated as the product of the number of concurrently used GPUs, the maximum rated GPU power, and a 95% utilization factor, which accounts for brief periods of non-GPU-intensive work. Based on Kyutai’s observations, we assume a CPU utilization of 5%. Following prior analyses of similar compute nodes (Spetko et al., 2020), we assume the power consumption of RAM and other node components (including fans, SSDs, network cards, and the motherboard) to be constant.
> >
> > [...]
> >
> > **Embodied Impacts.** We estimate embodied impacts for GPUs, CPUs, RAM, SSDs, power supplies, motherboards, and cases, as well as for the assembly of the compute nodes. Production and transport impacts of individual hardware components are estimated using per-component impact factors provided by Boavizta (Simon et al., 2025). For GPU production and transport impacts, we refer to a recent report by ADEME (Lees-Perasso et al., 2026)5. These embodied impacts are then allocated proportionally to the duration of hardware use during Moshi’s development relative to its typical service lifetime, following established practice in prior work (Luccioni et al., 2023; Morand et al., 2024; Falk et al., 2025).

---

> > ### Author Response · Authors · 2026-06-16
> > **Reply to reviewer K823 - 2**
> >
> > **After**
> > > [...] We distinguish between *operational impacts*, which correspond to the use phase of the hardware, and *embodied impacts*, which are associated with hardware production, transport, and end of life.
> > >
> > > **Operational impacts.** The main resource consumed during AI training is electricity, which indirectly results in water consumption through datacenter cooling. Additional operational impacts arise from the power plants generating this electricity.
> > >
> > > We first estimate GPU electricity consumption as the product of compute (GPU-time, defined as run duration multiplied by the number of GPUs used, aggregated across all runs), the maximum rated GPU power, and a GPU utilization factor. We set this utilization factor to 95%, based on observations from Kyutai indicating near-full GPU utilization during most runs, while accounting for brief periods of non-GPU-intensive work. However, these observations may not hold uniformly across all run types, particularly during under-optimized early experimentation or debugging runs. As a result, assuming a constant utilization factor close to full utilization for all runs may overestimate electricity consumption. We therefore provide a sensitivity analysis of this parameter in sec. C. `[Note to the reviewer: This sensitivity analysis can be found in the response to reviewer UMSy.]`
> > >
> > > We estimate the electricity consumption of CPUs, RAM, and the remaining node hardware in a similar manner, rescaling GPU-time as appropriate based on the quantity of hardware per compute node. Based on Kyutai’s observations, we assume a CPU utilization of 5%. Following prior analyses of similar compute nodes (Spetko et al., 2020), we assume the power consumption of RAM and other node components, including fans, SSDs, network cards, and the motherboard, to remain constant during training runs.
> > >
> > > [...]
> > >
> > > **Embodied Impacts.** Hardware manufacturing, transport, and end-of-life have direct impacts on the environment owing to rare mineral mining, ultrapure water consumption, fluorinated gas emissions, and more (Hess, 2024).
> > >
> > > We first estimate the unitary embodied impacts of GPUs, CPUs, RAM, SSDs, power supplies, motherboards, and chassis, as well as the assembly of the compute nodes. We then allocate embodied impacts to our functional unit, using the compute (GPU-time) associated with Moshi's development relative to the total hours of use of the hardware throughout its lifespan, following established practice in prior work (Luccioni et al., 2023; Morand et al., 2024; Falk et al., 2025). For components other than GPUs, we rescale compute based on the quantity of hardware per compute node.
> > >
> > > We estimate unitary embodied impacts using per-component impact factors provided by Boavizta (Simon et al., 2025). For GPU embodied impacts, we refer to a recent report by ADEME (Lees-Perasso et al., 2026).
> > >
> > > To estimate the total hours of use of the hardware, we assume a hardware lifespan of four years, in line with values employed in related work (Morand et al., 2024; Schneider et al., 2025; Falk et al., 2025; Desroches et al., 2025), and a reasonable average utilization rate of 60\% for all hardware components (Luccioni et al., 2023; Wu et al., 2022). We provide a sensitivity analysis of both parameters in sec. C. `[Note to the reviewer: This sensitivity analysis can be found in the response to reviewer UMSy.]`
> > >
> >
> >
> > *Additional reference: [Julia Christina Hess. Chip production’s ecological footprint: Mapping climate and environmental impact. Technical report, Interface (Stiftung Neue Verantwortung), 2024.](https://www.interface-eu.org/publications/chip-productions-ecological-footprint)*
> >
> > **Functional unit definition.** We provided an informal definition of our functional unit on page 11 of the submission, in the *Scope* paragraph:
> >
> > >The object of our assessment (or functional unit) covers the complete research and development process of the Moshi model, from early experiments to final trainings and ablations. We exclude data acquisition, processing, and storage due to a lack of detailed information, and we do not account for the environmental costs associated with deployment and inference after public release.
> >
> > We agree that this definition could be made more rigorous and explicit. We therefore revised the paragraph as follows:
> >
> > > The object of our assessment covers the complete research and development process of the Moshi model, from early experiments to final training runs and ablations. We exclude data acquisition, processing, and storage due to a lack of detailed information, and we do not account for the environmental costs associated with deployment and inference after public release. **Specifically, the *functional unit* of our assessment is: "Develop and release publicly a full-duplex speech-to-speech foundation model with a latency of 200 milliseconds".**

---

> > > ### Author Response · Authors · 2026-06-16
> > > **Reply to reviewer K823 - 3**
> > >
> > > **Methodological novelty.** Our contribution does not lie in proposing a new environmental modeling methodology, but in analyzing how compute is used during the research and development of a modern AI model, and relating this compute to environmental impacts using existing LCA methodologies. In particular, access to the complete development logs of Moshi allows us to quantify stages of model development that are typically omitted from prior analyses, including experimentation, debugging, ablation studies, and discarded runs. We additionally provide all methodological details in sec. B to facilitate reproducibility and encourage similar analyses in future work.
> > >
> > > **LCA framework specification.** One of the challenges we faced in this work is precisely the lack of a standardized LCA framework dedicated to supercomputing clusters and AI systems. As discussed in Section 2, several initiatives have recently emerged, but environmental assessment for AI remains an evolving field.
> > >
> > > In this work, we combine existing ICT-oriented LCA tools with modeling approaches proposed in prior studies on the environmental impacts of AI, which we adapt to our specific use case. Sec. B details how operational and embodied impacts are modeled, including all assumptions, parameters, and data sources used throughout the study. Section 2 also progressively introduces the terminology and scope commonly used in AI LCA studies, including first-, second-, and third-order impacts, as well as the different life-cycle phases of AI systems. We additionally clarified in the revised manuscript that our study follows an attributional LCA approach rather than a consequential one, a distinction now explicitly stated at the end of Section 2.1. We hope these additions clarify the framework and modeling choices underlying our analysis, and would be happy to further expand specific aspects if needed.
> > >
> > > **Choice of impact indicators.** Our choice of environmental impact indicators is primarily driven by the current limitations of the tools and data sources available for AI-focused LCA studies. In particular, the framework we use to estimate embodied impacts, Boavizta, currently supports only global warming potential, primary energy, and abiotic depletion potential (see, e.g., the [documentation for CPU impacts](https://doc.api.boavizta.org/Explanations/components/cpu/)). We encountered similar limitations for operational impacts: both existing tools and related work on AI environmental assessment typically focus on a restricted set of indicators, largely overlapping with those considered in our study (Morand et al., 2024).
> > >
> > > Our source for embodied GPU impacts does report a broader set of impact categories. However, we chose not to include these additional indicators because they would only apply to embodied GPU impacts, preventing any sort of comparison with other hardware or with operational impacts.
> > >
> > > More broadly, our objective is not to extend existing LCA frameworks for AI by introducing new impact indicators, but rather to provide a comprehensive analysis of the research and development costs of AI systems using the most consistently supported indicators currently available, while encouraging greater transparency and reproducibility in future reporting.

---

### Review · Reviewer_XrQp · 2026-06-07

**Summary Of Contributions:**

This paper is a measurement paper that studies the energy footprint of the Moshi model, which is a relatively novel speech-to-speech generative AI model. The authors analyze internal training logs, categorize compute across training phases and research activities, and estimate environmental impacts, including energy, carbon emissions, etc. The main message is that final training substantially underestimates the true environmental footprint of frontier-style GenAI research, because experimentation, failed runs, debugging, evaluation, and ablation studies dominate the total compute.

**Audience:**

Yes

**Audience Explanation:**

This paper provides interesting energy measurement results for training a novel generative AI model.

**Claims And Evidence:**

Yes

**Claims Explanation:**

The paper is a measurement paper, which essentially reports and records the entire training/development process. There are some approximations/assumptions for the energy measurements, and they all seem reasonable to me.

My main concern is generalization. Moshi is a relatively unusual case, as it is a novel speech-to-speech model developed largely from scratch by a new organization. This makes the case study interesting, but also limits how broadly one can interpret the headline result that final training accounts for only around 4% of total compute. The same ratio may not hold for more mature LLM pipelines, fine-tuning-heavy projects, derivative model releases, or organizations that reuse extensive prior infrastructure and recipes. The paper acknowledges some of this, but the conclusions and recommendations would benefit from more careful framing as a single-case study rather than a general characterization of Generative AI research.

Another problem is that again, this is a measurement paper, and it does not give solid ways to improve. The paper mentions some directions, such as reducing evaluation and safety analysis costs. However, I don't find such claims very solid. Firstly, they still don't contribute to a major amount of costs. Secondly, they may be the guardrails to prevent further costs in other categories, so it could even be the right strategy to increase such investments.

Finally, if possible, I think it would be very valuable to release the logs that are analyzed so that richer data and patterns could be identified, and be beneficial to the community.

**Requested Changes:**

Please check the above comments.

---

> ### Author Response · Authors · 2026-06-16
> **Reply to reviewer XrQp**
>
> We thank the reviewer for their interest in our work and for raising their generalization concerns. We agree that our phrasing should more clearly emphasize the case-study nature of our analysis, and we have revised the manuscript accordingly. The following paragraphs describe the corresponding changes made to the original text.
>
> **Generalization.** We agree that Moshi is relatively uncommon in the LLM landscape as a real-time speech-to-speech foundation model developed largely from scratch. As such, some of our results, particularly the finding that final training accounts for only around 4% of the total compute, should not be interpreted as universally representative of all AI development pipelines. Rather, our work provides an in-depth analysis of the complete research and development process of a frontier AI model, made possible by unprecedented access to the full internal training logs released by Kyutai.
>
> We therefore revised the discussion section to make this limitation more explicit. The beginning of Section 5 now reads as follows:
>
> **Before**
> > We conclude this study by discussing our main results, how they compare to those available in the literature, and possible mitigation strategies for the growing impact of Gen-AI research.
>
> **After**
> > We conclude this study by discussing our main results, how they compare to those available in the literature, and **good practices to adopt in GenAI research. Even though our results are specific to this case study and therefore cannot be directly generalized to any other AI research projects, the research stages we discuss (debugging, ablation studies, evaluation, etc.) are common to most AI development pipelines, and our findings suggest potential improvements of research practices.**
>
> Further changes in Section 5 are shown in our reply below. More generally, our goal is not to claim that the exact compute distribution observed for Moshi applies to all AI systems, but rather to demonstrate that important stages of AI development are typically omitted from environmental reporting because the required data is rarely available. We hope that this work encourages the release of similar reports for other systems, enabling broader comparisons and a better understanding of the environmental costs of AI research across different development settings.

---

> > ### Author Response · Authors · 2026-06-16
> > **Reply to reviewer XrQp - 2**
> >
> > **Improvement guidelines.** We appreciate the reviewer's insightful feedback. Our goal was not to claim universality, but rather to identify potentially important sources of environmental impact in AI research workflows and to suggest practical, actionable recommendations. We recognize that the original discussion section could have better reflected this nuance, and that some of our recommendations were phrased too strongly or simplistically. In response, we have revised Section 5 to address these concerns and improve clarity.
> >
> > In particular, we clarified that debugging and evaluation costs should not necessarily be minimized indiscriminately, as they can also help avoid substantially more expensive failed experiments or unstable training runs. The revised text now reads:
> >
> > **Before**
> > > **Reducing Unnecessary Compute.** [...] Debug runs, which are also inexpensive individually, still make up 2.4% of the research and development compute, almost as much as training the final model. Debugging is naturally necessary, but these figures invite to perform it as much as possible on infrastructures with a low power consumption instead of a production environment, potentially using downscaled versions of the models and datasets.
> > >
> > > Periodic evaluation and validation during model training account for 10% of the total compute. We believe that this important cost should be taken into account to modify standard practices, performing evaluation and validation at a lower frequency, and on smaller datasets.
> >
> > **After**
> > > **Reducing Unnecessary Compute.** [...] Debug runs, which are numerous but inexpensive individually, account for 3.9% (15 GPU-years) of the research and development compute in our case study, a share comparable to that of training the final model. Debugging is clearly essential, in particular because it can reduce overall compute usage by preventing failed large-scale training runs. Still, as our analysis shows, debug runs can accumulate into a non-negligible contribution. Researchers should therefore favor debugging on downscaled models and datasets whenever possible, and preferably use lower-power infrastructures rather than production environments. In most cases, debugging does not require access to frontier-scale hardware.
> > >
> > > Likewise, periodic evaluation and validation during training account for 10% of Moshi’s total compute, showing that performance tracking can induce a non-negligible computational cost. Periodic tracking during training can be important and has the potential to reduce overall compute usage by, for example, enabling early stopping or the diagnosis of unstable training runs, which can then be promptly terminated. Thus, considering the trade-off between potential benefits and costs of periodic evaluation, and limiting the compute spent on evaluation (e.g. keeping it under a small percentage of the compute of a run), seems important. In many settings, this cost could be reduced by lowering evaluation frequency and using smaller validation sets.
> >
> > **Before**
> > > **Questioning Research Practices and Expectations.** Ablation studies are often at the core of a machine learning research article, validating the findings and claims. However, these ablation studies have an important cost: 8% of the compute of Moshi being spent on ablation studies and safety analyses, mostly carried out while the final model was already trained. This relatively large share stems mainly from ablations on pre-training. We believe that this should lead to more questioning of the necessity and practices of ablation studies. For example, most comparisons could be carried out on smaller versions of the models and datasets or after much fewer training iterations. [...]
> >
> > **After**
> > > **Questioning Research Practices and Expectations.** Ablation studies are often central to machine learning research articles, as they validate findings and methodological claims. However, they can also induce substantial computational costs: in our case study, 11% of Moshi’s total compute was spent on ablation studies and safety analyses, mostly after the final model had already been trained. This share is roughly three times larger than the compute required to train the final deployed model itself, and stems primarily from expensive pre-training ablations.
> > >
> > > We believe these findings encourage giving more care to ablation selection and design. For example, comparisons could preferentially be carried out at earlier stages of training, through short fine-tuning runs, or systematically on smaller versions of the models and datasets. This also encourages an evolution of reviewers' mindsets, who should likewise remain mindful of the computational costs associated with expected ablations or analyses. [...]

---

> > > ### Author Response · Authors · 2026-06-16
> > > **Reply to reviewer XrQp - 3**
> > >
> > > The key takeaways in Section 3.1 are specific to Moshi, but we have also modified them to mitigate overly-generic claims:
> > >
> > > **Before**
> > > > * **Periodic evaluation during training adds a significant overhead**: over 7% of the compute is spent on evaluating models in case the need for deeper analysis or human assessment arises, which calls into question the common practice of performing these evaluations regularly, and invites to consider using less expensive and less frequent performance tracking.
> > > > [...]
> > > > * **Ablation studies are costly**: while they are key to rigorous design choice validation and research publications, ablation studies represent 8% of the total computation budget, again questioning common research practices.
> > >
> > > **After**
> > > > * **Periodic evaluation during training adds a significant overhead**: over 7% of the compute is spent on evaluating models in case the need for deeper analysis or human assessment arises, **a non-negligible share that calls into question the need for additional extensive online evaluation on top of validation.**
> > > > [...]
> > > > * **Ablation studies are costly**: while they are key to rigorous design choice validation and research publications, ablation studies represent 11% of the total computation budget, **a considerable share that invites to consider alternative design validations.**
> > >
> > > Please let us know if there are any remaining claims that you find concerning.
> > >
> > > **Data release.** We have obtained authorization from Kyutai to release an anonymized version of the training logs: https://anonymous.4open.science/r/GenAIFootprint-0C9C. The released data include, for each run, the number of requested GPUs; the training, validation, evaluation, and generation durations; the creation and modification times of the log entry; a training phase identifier; and tags for run classification (debug, final, ablation, etc.). We compliment the data file with metadata explaining the meaning of each field and how it was derived, if applicable.
> > >
> > > Example of a log entry:
> > >
> > > ```
> > > {
> > >     "id": 57,
> > >     "gpus": 2,
> > >     "durations": {
> > >         "train": 2.9230795332,
> > >         "valid": 0.2250824352,
> > >         "evaluate": 0.6980337475,
> > >         "generate": 0.0022963946
> > >     },
> > >     "ctime": 1702994663,
> > >     "mtime": 1702883370,
> > >     "phase": "A",
> > >     "tags": "debug"
> > > }
> > > ```
> > >
> > > **Note:** while preparing the data for release, we refined the preprocessing of the raw logs, in particular by improving the removal of duplicate experiment entries. This led to the reclassification of a small number of experiments, resulting in minor updates to Figure 4. The remaining figures and conclusions are essentially unchanged, and the environmental assessment is unaffected by these modifications.

---

### Decision · Action_Editor_SCyj · 2026-07-10

**Recommendation:** Accept with minor revision

**Additional Comments:**

In the current discussion section, there is a clear caveat about the specificity of this case study:
"Even though our results are specific to this case study and therefore cannot be directly generalized to any other AI research projects, the research stages we discuss (debugging, ablation studies, evaluation, etc.) are common to most AI development pipelines, and our findings suggest potential improvements of research practices."
I would like the authors to make this caveat clearer in the intro and the abstract (more briefly in the latter).

**Audience:**

Yes

**Audience Explanation:**

The environmental impacts of generative AI have recently been a topic of broad public and within-field discussion; the reviewers generally agreed that some readers would be interested in this paper, and I concur.

**Claims And Evidence:**

Yes

**Claims Explanation:**

The reviewers agree that this paper supports its claims, particularly after the revision, though one reviewer continued to be concerned about overstating the generality of the findings from this case study, the paper does caveat the specificity in the beginning of the discussion very clearly. In preparing the camera ready, I would like the authors to be a bit more explicit about this in the abstract and introduction as well.